# PROCEEDINGS A

applied mathematics, mathematical modelling, biophysics

parameter estimation, uncertainty quantification, stochastic model, random walk

**Author for correspondence:**
Matthew J. Simpson
e-mail: matthew.simpson@qut.edu.au

# Profile likelihood analysis for a stochastic model of diffusion in heterogeneous media

Matthew J. Simpson[1], Alexander P. Browning[1], Christopher Drovandi[1], Elliot J. Carr[1], Oliver J. Maclaren[2] and Ruth E. Baker[3]

[1]School of Mathematical Sciences, Queensland University of Technology, Brisbane, Australia
[2]Department of Engineering Science, University of Auckland, Auckland 1142, New Zealand
[3]Mathematical Institute, University of Oxford, Oxford OX2 6GG, UK

MJS, 0000-0001-6254-313X; CD, 0000-0001-9222-8763;
EJC, 0000-0001-9972-927X; REB, 0000-0002-6304-9333

We compute profile likelihoods for a stochastic model of diffusive transport motivated by experimental observations of heat conduction in layered skin tissues. This process is modelled as a random walk in a layered one-dimensional material, where each layer has a distinct particle hopping rate. Particles are released at some location, and the duration of time taken for each particle to reach an absorbing boundary is recorded. To explore whether these data can be used to identify the hopping rates in each layer, we compute various profile likelihoods using two methods: first, an exact likelihood is evaluated using a relatively expensive Markov chain approach; and, second, we form an approximate likelihood by assuming the distribution of exit times is given by a Gamma distribution whose first two moments match the moments from the continuum limit description of the stochastic model. Using the exact and approximate likelihoods, we construct various profile likelihoods for a range of problems. In cases where parameter values are not identifiable, we make progress by re-interpreting those data with a reduced model with a smaller number of layers.

# 1. Introduction

Mathematical models of biological phenomena are essential tools that can be used to improve our understanding and interpretation of experimental data and underlying mechanisms [1]. Methods for *identifiability analysis* allow us to objectively determine if available biological data are sufficient to estimate model parameters [2,3]. A mathematical model is said to be identifiable when distinct parameter values imply distinct distributions of observations. This in turn means parameters can be recovered from full knowledge of these distributions [4]. In the systems biology literature, identifiability is also referred to as *structural identifiability* where we consider an infinite amount of ideal, noise-free data [5–9]. By contrast, the term *practical identifiability* describes whether it is possible to provide reasonably precise parameter estimates using finite amounts of non-ideal, noisy data [10–13]. Practical identifiability can be assessed using both Bayesian [11,14] and frequentist methods, where the latter is commonly carried out using profile likelihood analysis [15,16]. Profile likelihood is often more computationally efficient [17], and also tends to perform better in the presence of true structural non-identifiability [13].

In the systems biology literature, experimental data often take the form of time series that are modelled using systems of ordinary differential equations [18]. A profile likelihood analysis can be undertaken by making assumptions about both the process model (e.g. a system of ordinary differential equations) and a noise model (e.g. Gaussian noise with zero mean and constant variance) [10,11]. Together, these define a likelihood function, and numerical optimization tools can be used to compute various profile likelihoods that provide rapid insight into identifiability. For conditions where parameters are identifiable, the profile likelihood can be used to obtain maximum-likelihood point estimates, as well as confidence intervals by choosing a threshold-relative profile likelihood value [10]. Most profile likelihood analyses in the mathematical biology literature have focused on deterministic process models, such as models based on ordinary differential equations [10] and partial differential equations [17]. While some studies have considered profile likelihood analysis of stochastic models [19,20], here we focus on computing profile likelihoods based on first-exit time data from a stochastic model of diffusion in layered media that is implemented to mimic a very practical experimental scenario that places tight restrictions on how data are observed, which we will now explain.

We consider a stochastic model that describes the spatio-temporal diffusion of thermal energy in a heterogeneous, layered biological material. We are motivated by the experimental work of Andrews *et al.* [21–24] who consider heat conduction through living, layered porcine skin, shown in figure 1*a*. Andrews' experiments are performed by placing a constant temperature external heat source at the surface of the skin, at the top of the epidermis. Over time, thermal energy conducts across the epidermis and dermis, reaching the subdermal fat layer. Andrews *et al.* [21–24] measure this heat conduction process in real time by obliquely inserting a temperature probe at the base of the fat layer. For example, the data in figure 1*b* show the result of an experiment where a constant heat source at 50°C is placed at the top of the skin, and the subdermal temperature is measured at intervals of 1 s at the base of the fat layer. The recorded data show that the subdermal temperature remains approximately constant for the first 14 s of the experiment before increasing with time. These data suggest that the thermal disturbance at the surface of the layered skin takes approximately 14 s to affect the subdermal temperature. A key restriction of Andrews' experimental design is that the living tissues are relatively thin and the temperature probe can only be placed at a single location without destroying the integrity of the living tissue [21–24]. Given that Andrews' data take the form of a time series of temperature data recorded at the base of the fat layer, a key quantity of interest is to measure the duration of time required for the temperature at the base of the fat layer to respond to the thermal disturbance at the surface of the layered skin. While Andrews' experiments focus on diffusive transport of thermal energy in layered biological material [21–24], a very similar experimental design could be used to measure the chemical diffusion of dissolved solutes through skin and other heterogeneous, layered media, with broad applications including cutaneous drug delivery [25] as well as the design of landfill liners for the storage of industrial waste [26,27].

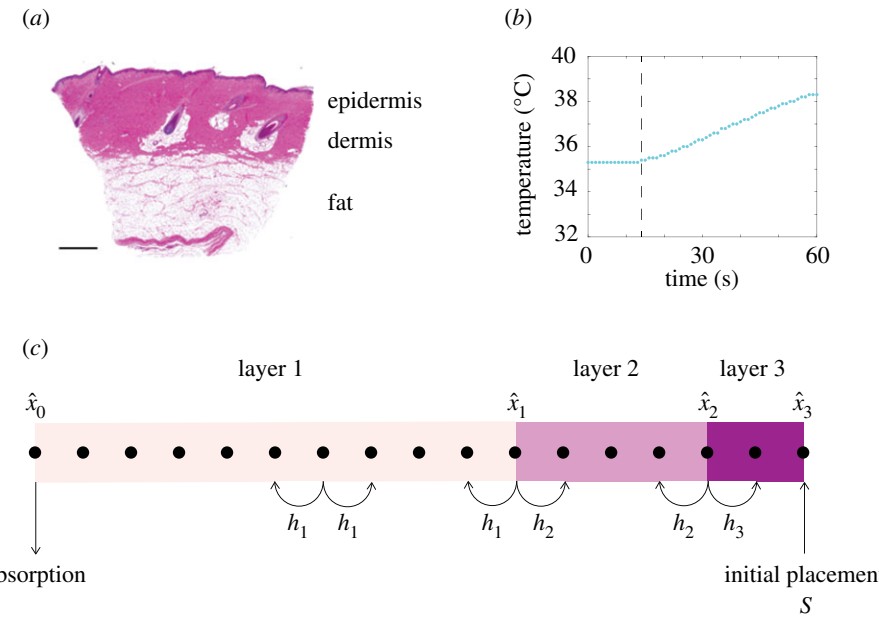

**Figure 1.** (*a*) Histology image of porcine skin [23] highlighting the layered structure of the tissue, including the epidermis, dermis and fat layers, as indicated. (*b*) Experimental data from Andrews *et al.* [21] showing the subdermal temperature response after a constant heat source at 50°C is placed on the surface of the skin. Subdermal temperatures are measured at 1 s intervals, and it is clear that the thermal response at the surface takes approximately 14 s to reach the subdermal probe, as indicated by the vertical line. (*c*) Schematic of the discretized domain with $m = 3$ layers and $n = 17$ lattice sites. Data in (*b*) are available from Andrews *et al.* [21]. (Online version in colour.)

Some of our previous modelling work attempted to calibrate the solution of deterministic partial differential equation models of heat transfer to match the kind of data reported by Andrews *et al.* [21,22]. In this previous work, we treated the layered skin in figure 1*a* as a very simple two-layer problem, where we took the thermal diffusivity of the combined epidermis and dermis to be $D_1 > 0$, and the thermal diffusivity of the fat layer to be $D_2 > 0$ [24]. Using this very simple approach, we showed that it is impossible to estimate $D_1$ and $D_2$ using time-series measurements of temperature at one location since multiple combinations of $D_1$ and $D_2$ give rise to the same time-series temperature signal at the base of the material (see, for example, fig. 2*c,d* in McInerney *et al.* [24]). A key question that remains unanswered by the work of McInerney *et al.* [24] is whether our ability to identify thermal diffusivities using these kind of observations can be improved by working with a stochastic, rather than a deterministic model. To address this question, here we mimic the kind of data reported by Andrews *et al.* [21] using a simple stochastic random walk model of heat conduction in a one-dimensional layered material. We do not precisely mimic the exact layer thicknesses, boundary conditions, initial conditions and temperature data reported by Andrews *et al.* [21]. Instead, we address a more fundamental question of exploring whether working in a stochastic modelling framework offers additional insights over working with simpler deterministic models. We address this question by working with idealized boundary conditions and initial conditions that do not precisely capture the details of Andrews' experiments. However, even with this highly idealized modelling framework we show that estimating diffusivities in layered materials with simple time-series data at one location is extremely challenging. Indeed, we show that it is only possible to reliably estimate these parameters by changing the experimental design. Since this work focuses on stochastic methods, which are more computationally expensive than deterministic approaches, a major question we address is the development of methodological tools that speed up identifiability analysis through the development of an approximate likelihood. The approximations we develop and deploy are novel, leading to significant computational savings, without sacrificing accuracy.

We mimic Andrews' experimental design using a stochastic random walk model of diffusion in layered, heterogeneous media [28–30]. In this model, particles undergo a one-dimensional random walk across a series of internal layers, where each layer has a different particle hopping rate, shown schematically in figure 1c. To be consistent with Andrews' experiments, in the first instance, we consider releasing particles at a single location, such as the end of the domain that represents the surface of the tissue, and the simulation proceeds until the particle reaches an absorbing boundary at the other end of the domain, that represents the base of the tissue. Simulations are summarized by recording the duration of time taken for the particle to reach the absorbing boundary. Our aim is to understand if such data are sufficient to characterize the hopping rates, and hence the thermal diffusivity, of each layer. To analyse the practical identifiability of this model with these data, we construct profile likelihoods using two approaches. First, we interpret the stochastic model as a Markov chain and compute an exact likelihood from which we can compute various profile likelihoods. Second, we suppose the likelihood can be approximated by a Gamma distribution where the first two moments are defined in terms of properties of the stochastic model. Given this approximate likelihood we then construct various profile likelihoods using numerical optimization. The first approach is exact but relatively expensive, whereas the second approach is approximate but computationally fast. Our results show that the approximate likelihood approach provides reasonably accurate, rapid insight into parameter identifiability. In cases where the hopping rates are not identifiable, we make progress on parameter estimation by either refining the experimental design to provide more data, or by introducing homogenization approximations that allow us to consider a reduced model with fewer parameters.

## 2. Results and discussion

### (a) Stochastic model

We consider a stochastic random walk on an interval $[0, L]$ that is partitioned into $m$ non-overlapping layers $(\hat{x}_{i-1}, \hat{x}_i)$ for $i = 1, 2, \ldots, m$, where $0 = \hat{x}_0 < \hat{x}_1 < \hat{x}_2 < \ldots < \hat{x}_{m-1} < \hat{x}_m = L$ are the locations of the interfaces, and $\hat{x}_i$ is the location of the interface between layer $i$ and layer $i + 1$, as shown in figure 1c. To simulate a random walk on this domain, we discretize the interval $[0, L]$ uniformly using a unit lattice, $\Delta = 1$, giving $n = L + 1$ lattice sites. Sites are indexed so that the position of site $j$ is $x_j = \Delta(j - 1)$ for $j = 1, 2, \ldots, n$. The positions of the interfaces are chosen so they coincide with a lattice site. This means that we have $n$ lattice sites, indexed $j = 1, 2, \ldots, n$, and $m$ layers indexed $i = 1, 2, \ldots, m$. Typically, we consider applications where $n \gg m$ so that we have many more lattice sites than layers, as in figure 1c.

A particle is placed onto the lattice at position $S$ and undergoes an unbiased random walk. Time is uniformly discretized into intervals of unit duration, $\tau = 1$. When a particle is located within layer $i$, during the next time step of duration $\tau$, the particle either: (i) hops left with probability $h_i \leq 0.5$; (ii) hops right with probability $h_i \leq 0.5$; or (iii) remains stationary with probability $1 - 2h_i$. When a particle is located at an interface $\hat{x}_i$, during the next time interval of duration $\tau$, the particle either: (i) hops left with probability $h_i \leq 0.5$; (ii) hops right with probability $h_{i+1} \leq 0.5$; or (iii) remains stationary with probability $1 - h_i - h_{i+1}$. An absorbing boundary condition is applied at $j = 1$ and a reflecting boundary condition is applied at $j = n$. This means that a particle located at site $j = n$ is unable to hop to the right, and a particle that hops to site $j = 1$ is removed, and the simulation stops at that point where we record the duration of time required for the particle to be captured at the absorbing boundary. When we consider an ensemble of identically prepared realizations we obtain a distribution of exit times, and we can interpret the mean of that distribution as a measure of the duration of time required for some kind of disturbance to diffuse across the spatial domain [31,32]. Here, we use the most straightforward stochastic modelling approach where we consider discrete space and discrete time. However, other approaches, such as discrete space continuous time, or continuous space discrete time modelling approaches are also valid approaches.

Motivated by the experiments reported by Andrews *et al.* [21], we consider simulations for a fixed geometry where the number of layers, $m$, and the location of the interfaces, $\hat{x}_0, \hat{x}_1, \ldots, \hat{x}_m$, are treated as known, measurable constants. Our aim is to determine whether the hopping rates in the different layers, $h_1, h_2, h_3, \ldots, h_m$, can be estimated by observing exit time data from a suite of random walk simulations. An individual simulation is performed by releasing a particle at a particular location, $S$. Each simulation is performed until the particle leaves the system at the absorbing boundary after some duration of time. This exit time is related to the measurements reported by Andrews *et al.* [21] since they are particularly interested in the time taken for the thermal energy to diffuse across the layered skin. To perform a simulation we must specify several parameters, $\phi = (h_1, h_2, \ldots, h_m, S)$. Compared to the interface locations, $S$ is a parameter in the sense that we can choose different values of $S$ whereas we treat the positions of the layer interfaces as fixed, measurable quantities [21]. Therefore, the free parameters that we wish to estimate are $\theta = (h_1, h_2, \ldots, h_m)$. In this work, we first consider releasing particles at the reflecting boundary, $S = L$, and those particles are captured at the absorbing boundary, $x = 0$. This straightforward design reflects the geometry of Andrews *et al.*'s [21] experiments illustrated in figure 1. Subsequently, we explore the effect of varying the choice of release point, $S$, and later we also consider other designs by releasing particles at more than one initial location to explore how this affects parameter identifiability. While placing heat sources at different depths is not possible in Andrews *et al.*'s experiments [21], it is of theoretical interest to explore how parameter identifiability is affected by releasing particles at more than one location.

Denote $T \sim F(t \mid \theta)$ as the random variable describing the exit time with cumulative distribution function $F(t \mid \theta)$ and corresponding probability density function $f(t \mid \theta)$, where $t$ is a realization of $T$ for fixed $S$. These distributions depend upon the hopping rates in each layer, $\theta = (h_1, h_2, \ldots, h_m)$. We further assume the data available for analysis consist of $R$ independent, identically prepared experiments, producing a random sample $\mathbf{t} = (t_1, t_2, \ldots, t_R)$ drawn from $T$. After observing such an ensemble of experiments we can visualize the exit time data as a histogram, and we can summarize the data in terms of the moments of exit time.

The key question we address here can be stated as follows: given a particular geometry, where $\hat{x}_0, \hat{x}_1, \ldots, \hat{x}_m$ are known, together with some observation of the exit time data, can we estimate the transport coefficients, $h_i$ for $i = 1, 2, 3, \ldots, m$, and what is the uncertainty in these estimates? Motivated by Andrews *et al.* [21] we first consider a three layer problem, $m = 3$, with $\hat{x}_0 = 0$, $\hat{x}_1 = 30$, $\hat{x}_2 = 60$ and $\hat{x}_3 = L = 100$, with free parameters $\theta = (h_1, h_2, h_3)$. An ensemble of $R = 10^4$ simulations, each with $S = 100$, is summarized in terms of the distribution of exit times in figure 2. Since all particles are released at $S = L = 100$, and are eventually absorbed at $x = 0$, every particle passes through each layer and so the exit time is influenced by the hopping rates in each layer. While, in principle, we could generate many ensembles for different $\theta$ and compare the resulting histograms, this approach would be extremely computationally expensive owing to the stochastic nature of the model. A much simpler approach that we will follow is to instead summarize the histogram in terms of a parametric distribution. In this particular example, we also have the ability to compute the exact exit time distribution by analysing the random walk as a Markov chain. This gives us the ability to compare the accuracy of the approximation.

## (b) Exact analysis

Given the independence of the $R$ exit times, the likelihood function is a product of the individual exit time likelihoods, and so we focus on discussing how to compute or approximate $f(t \mid \theta)$ for a single realization $t$. In this section, we discuss how to compute the probability density function, and in §2c how it can be approximated in a computationally efficient manner.

In the discrete model, we have a reflecting boundary at $x = L$, so that attempts to move in the positive $x$ direction from $x = L$ are aborted, and the absorbing boundary at $x = 0$ so that particles are removed once they reach $x = 0$. The probabilities of moving left and right from site $j$ in the

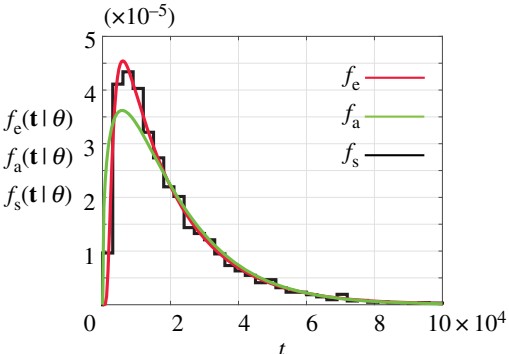

**Figure 2.** Simulation data for a three-layered problem with $m = 3, \hat{x}_0 = 0, \hat{x}_1 = 30, \hat{x}_2 = 60$ and $\hat{x}_3 = L = 100$. Simulations correspond to $\theta = (0.2, 0.3, 0.4)$ and $S = L = 100$, with an absorbing boundary condition at $x = 0$ and a reflecting boundary condition at $x = L$. The black histogram shows exit time data, $f_s(\mathbf{t} \mid \theta)$, constructed using $R = 10^4$ identically prepared simulations of the stochastic model. The red curve is $f_e(\mathbf{t} \mid \theta)$, obtained by treating the random walk as a Markov chain. The green curve is the approximate exit time distribution given by the Gamma distribution, $f_a(\mathbf{t} \mid \theta)$. Here, we have $a = 1.40$ and $b = 1.45 \times 10^4$, which matches the first two moments of exit time in the continuum limit. (Online version in colour.)

three-layer problem are given by

$$\mathbb{P}_{l,j} = \begin{cases} h_1, & j = 2, 3, \ldots, \hat{x}_1, \\ h_2, & j = \hat{x}_1 + 1, \hat{x}_1 + 2, \ldots, \hat{x}_2, \\ h_3, & j = \hat{x}_2 + 1, \hat{x}_2 + 2, \ldots, \hat{x}_3 \end{cases} \tag{2.1a}$$

and

$$\mathbb{P}_{r,j} = \begin{cases} h_1, & j = 2, 3, \ldots, \hat{x}_1 - 1, \\ h_2, & j = \hat{x}_1, \hat{x}_1 + 1, \ldots, \hat{x}_2 - 1, \\ h_3, & j = \hat{x}_2, \hat{x}_2 + 1, \ldots, \hat{x}_3 - 1, \\ 0, & j = \hat{x}_3. \end{cases} \tag{2.1b}$$

Denote $X(t_k) \in \{1, \ldots, L\}, k = 0, 1, 2, \ldots$, as a sequence of random variables describing the location of a particle at time $t_k$ given that it was initially located at $S$. Here, $k$ is an integer that denotes the number of time steps and $t_k = \tau k$ is the discrete time value. The sequence $X(t_k)$ forms a time-homogeneous Markov chain. We denote by $p(x, t_k)$ the probability mass function of $X(t_k)$ where $x \in \{1, \ldots, L\}$ denotes a specific location. Denoting $\mathbf{p}(t_k) = (p(1, t_k), p(2, t_k), \ldots, p(L, t_k))^\top$, then $\mathbf{p}(t_k)$ can be obtained using standard results from Markov chain theory,

$$\mathbf{p}(t_k) = \mathbf{P}^k \cdot \mathbf{p}(0), \tag{2.2}$$

were $\mathbf{P}$ is the $L \times L$ tridiagonal transition matrix, with upper, main and lower diagonal entries $\mathbf{P}_u = (\mathbb{P}_{l,2}, \mathbb{P}_{l,3}, \ldots, \mathbb{P}_{l,L})$, $\mathbf{P}_d = (1 - \mathbb{P}_{l,1} - \mathbb{P}_{r,1}, \ldots, 1 - \mathbb{P}_{l,L} - \mathbb{P}_{r,L})$, and $\mathbf{P}_l = (\mathbb{P}_{r,1}, \mathbb{P}_{r,2}, \ldots, \mathbb{P}_{r,L-1})$, respectively, and $\mathbf{p}(0)$ is a vector of zeros, except for the $X$th element, which takes the value of unity. With these definitions, the cumulative distribution function of the exit time distribution is

$$F(t \mid \theta) = p(1, t). \tag{2.3}$$

Here, the definition of the cumulative distribution function has a very straightforward interpretation. Simulations continue until the particle is absorbed at $j = 1$. Therefore, $p(1, t)$ is a measure of the probability that the particle is captured, or accumulates, at the absorbing boundary during the $t$th time step. Given this definition, the associated probability mass function, $f_e(t \mid \theta)$, is

$$f_e(t \mid \theta) = \begin{cases} 0, & t = 0, \\ F(t \mid \theta) - F(t - 1 \mid \theta), & t = 1, 2, \ldots, \end{cases} \tag{2.4}$$

where the subscript in $f_e(t \mid \theta)$ denotes that this is an *exact* formulation for the probability mass function, which gives the probability that the particle is absorbed during the $t$th time step.

In practice, we compute $f_e(t \mid \theta)$ by computing $\mathbf{P}^t$ and $\mathbf{P}^{t-1}$ by diagonalizing the transition matrix. The diagonalization of the transition matrix allows us to compute $f_e(t \mid \theta)$ efficiently, and it turns out that this computation of $f_e(t \mid \theta)$ is faster than considering a sufficiently large number of identically prepared realizations of the stochastic model. However, we still find that the Markov chain computation is considerably more expensive relative to the approximations that we will develop in §2c. To check our results, we compute the exact exit time probability mass function for the three-layer problem described in figure 2, and compare $f_e(t \mid \theta)$ with data from $R = 10^4$ stochastic realizations in figure 2 where we see that the exact Markov chain calculation matches the simulation data very well. Of course, the comparison between $f_e(t \mid \theta)$, $f_a(t \mid \theta)$ and $f_s(t \mid \theta)$ in figure 2 is for one particular example of a three-layered problem for a particular choice of $\theta$; however, additional results (not shown) indicate that we obtain a similarly good match between these quantities for other problems with different $m$ and $\theta$.

## (c) Approximate analysis

To proceed with deriving an approximate likelihood, we make the natural assumption that the exit time is distributed according to a Gamma distribution with density $\Gamma(t; a, b)$ for realization $t$, with mean and variance $a/b$ and $a/b^2$, respectively. This amounts to assuming that the time the particle spends in each layer is exponentially distributed, so that the sum of several exponentially distributed events gives rise to an exit time distribution that is approximated as a Gamma distribution. To demonstrate, we calculate the first two central moments of the exit time distribution and compare it to the associated Gamma distribution, $\Gamma(t; a, b)$, in figure 2. This approximation gives a reasonable qualitative match with the simulated data and the exact distribution. The main point here is that the Gamma distribution is a simple parametric distribution that can be evaluated very cheaply. By contrast, the simulated and exact distributions require far more computational effort, and the amount of computational effort depends upon the geometry of the problem and the choice of parameter values. As in the case of many stochastic models [33], we can formulate algebraic expressions for the moments in terms of the model parameters, which allows us to approximate the exit time distribution, $f_a(t \mid \theta) = \Gamma(t; a, b)$, by matching the first two moments to the Gamma distribution. To implement this approximation, we now explain how to calculate the moments without using stochastic simulations.

We can analyse the stochastic model in terms of the moments of exit time by using ideas related to the first passage time [29,34–36]. To make progress, we analyse the stochastic model in the continuum limit by treating the starting location, $S$, as a continuous variable $x$ on $0 < x < L$. In the usual way, the stochastic model is associated with a macroscopic diffusivity $D_i = \Delta^2 2h_i/2\tau$, in the limit that $\Delta \to 0$ and $\tau \to 0$ jointly, as the ratio $\Delta^2/\tau$ remains finite [29,34–36]. This means that we have $D_i = h_i$ in our non-dimensional framework with $\Delta = \tau = 1$. Of course, this non-dimensional model can be applied to any dimensional problem by re-scaling $\Delta$ and $\tau$ as appropriate [37].

The $n$th raw moment of exit time for a particle released at $x$ is governed by a system of linear boundary value problems [29,30],

$$D_i \frac{d^2 M_n^{(i)}(x)}{dx^2} = -M_{n-1}^{(i)}(x), \quad \hat{x}_{i-1} < x < \hat{x}_i, \tag{2.5}$$

for each layer $i = 1, 2, \ldots, m$, and for each moment $n = 1, 2, \ldots$, with $M_0^{(i)}(x) = 1$ for each layer $i = 1, 2, \ldots, m$. To solve for the $n$th moment, we specify boundary conditions

$$M_n^{(1)}(0) = 0, \quad \frac{dM_n^{(m)}(L)}{dx} = 0, \tag{2.6}$$

and to close this system we require additional conditions at each interface location:

$$M_n^{(i)}(\hat{x}_i) = M_n^{(i+1)}(\hat{x}_i), \quad -D_i \frac{dM_n^{(i)}(\hat{x}_i)}{dx} = -D_{i+1} \frac{dM_n^{(i+1)}(\hat{x}_i)}{dx}. \tag{2.7}$$

The solution of equation (2.5) for the first raw moment, $M_1(x)$, is a piecewise quadratic function of position within each layer, $M_1^{(i)}(x)$ for $i = 1, 2, 3, \ldots, m$. Applying the conditions (2.6)–(2.7) determines the coefficients of these quadratic functions. For an arbitrary layer arrangement, the first raw moment is given by

$$M_1^{(i)}(x) = \sum_{k=1}^{i-1} \frac{\ell_k^2}{2} \left[ \frac{1}{D_k} - \frac{1}{D_i} \right] + \sum_{k=1}^{i-1} \sum_{j=k+1}^{m} \ell_k \ell_j \left[ \frac{1}{D_k} - \frac{1}{D_i} \right] + \frac{x(2L - x)}{2D_i}, \quad \hat{x}_{i-1} < x < \hat{x}_i, \quad (2.8)$$

for $i = 1, 2, 3, \ldots, m$, and $\ell_i = \hat{x}_{i+1} - \hat{x}_i$ is the length of the $i$th layer. The solutions of equation (2.5) for the second and higher raw moments are higher order polynomials of position. For an arbitrary number of layers these solutions rapidly become algebraically complicated for $n \geq 2$. However, these expressions can be evaluated easily using symbolic software that is available on GitHub (https://github.com/ProfMJSimpson/StochasticIdentifiability).

Given the geometry of the problem, $\hat{x}_0, \hat{x}_1, \ldots, \hat{x}_m$, and the hopping probabilities, $\theta = (h_1, h_2, \ldots, h_m)$, we can calculate the first two raw moments, $M_1(x)$ and $M_2(x)$, and convert these into the mean and variance, $T(x) = M_1(x)$ and $V(x) = M_2(x) - M_1(x)^2$, respectively. This information allows us to approximate the distribution of exit times for a particle released at $x$, $f(t \mid \theta) \approx f_a(t \mid \theta) = \Gamma(a, b)$, with $T(x) = a/b$ and $V(x) = a/b^2$. This approach is computationally efficient since it completely avoids performing stochastic simulations or Markov chain calculations. In the electronic supplementary material S1, figure S1, we compare appropriately averaged data from the stochastic model with the solution of the continuum expressions for the moments of exit time.

Our approximate method is a form of indirect inference (e.g. [38]), a method originally developed in econometrics for parameter estimation of intractable likelihood models. The approach involves selecting an auxiliary model with a tractable likelihood that is not designed to explain how the data were generated, but can still provide a reasonable description of the data. Denote the auxiliary likelihood as $f_A(\mathbf{t}|\phi)$ where $\phi$ is the parameter of the auxiliary model and we must have $\dim(\phi) \geq \dim(\theta)$. The main objective of indirect inference is to establish a mapping between the auxiliary parameter $\phi$ and parameter of interest $\theta$; abusing the notation we define this as $\phi(\theta)$. In our application, $\phi$ represents the parameters of the Gamma distribution. Fortunately, in our case, we have a method for directly computing the mapping. Following [38], we approximate the likelihood as $f(\mathbf{t}|\theta) \approx f_A(\mathbf{t}|\phi(\theta))$. More information on the form of the likelihood for our model is provided in the next section.

## (d) Likelihood function

Given a suite of $R$ identically prepared observations associated with a particular geometry, $\hat{x}_0, \hat{x}_1, \ldots, \hat{x}_m$, a particular release point, $S$, and hopping rates, $\theta = (h_1, h_2, \ldots, h_m)$, we can now develop expressions for the *exact* and *approximate* likelihood functions. Denoting the full observed dataset as $\mathbf{t} = (t_1, \ldots, t_R)$ the exact likelihood function is given by

$$\mathcal{L}_e(\theta \mid \mathbf{t}) = \prod_{r=1}^{R} f_e(t_r \mid \theta), \quad (2.9)$$

where $f_e(t_r \mid \theta)$ is the exact probability mass function for the $r$th observed exit time. An exact expression for the log-likelihood is then

$$\ell_e(\theta \mid \mathbf{t}) = \sum_{r=1}^{R} \ln \left( f_e(t_r \mid \theta) \right). \quad (2.10)$$

Similarly, the approximate likelihood function is given by

$$\mathcal{L}_a(\theta \mid \mathbf{t}) = \prod_{r=1}^{R} f_a(t_r \mid \theta). \quad (2.11)$$

Here, $f_a(t_r \mid \theta) = \Gamma(t_r; a, b)$ is the approximate probability mass function for the $r$th observed exit time, where $T(x) = a/b$ and $V(x) = a/b^2$. An expression for the approximate log-likelihood is

$$\ell_a(\theta \mid \mathbf{t}) = \sum_{r=1}^{R} \ln \left( f_a(t_r \mid \theta) \right). \tag{2.12}$$

In equations (2.10) and (2.12), we write the summation from $r = 1$ to $r = R$ which we can interpret as releasing $R$ particles at one particular location, $S$. Later, we will compare estimates where we release particles at multiple locations. Since the particle trajectories are all independent, we compute the exact and approximate log-likelihoods in the same way, by summing over the total number of particles released.

## (e) Profile likelihood

Here, we describe how profile likelihood identifiability analysis can be undertaken. We will describe the process in terms of the exact log-likelihood function, $\ell_e(\theta \mid \mathbf{t})$, but the analogous calculations can be performed with the approximate log-likelihood function, $\ell_a(\theta \mid \mathbf{t})$. These definitions are based on the same log-likelihood function $\ell_e(\theta \mid \mathbf{t})$, but here we will present results in terms of the normalized log-likelihood function, denoted

$$\hat{\ell}_e(\theta \mid \mathbf{t}) = \ell_e(\theta \mid \mathbf{t}) - \sup_{\theta} \ell_e(\theta \mid \mathbf{t}), \tag{2.13}$$

which we consider as a function of $\theta$ for fixed data, $\mathbf{t}$.

We assume our full parameter $\theta$ can be partitioned into an *interest* parameter $\psi$ and *nuisance* parameter $\lambda$, i.e. $\theta = (\psi, \lambda)$. Given a set of exit time data, $\mathbf{t}$, the profile log-likelihood for the interest parameter $\psi$ can be written as [2,39]

$$\ell_p(\psi \mid \mathbf{t}) = \sup_{\lambda} \hat{\ell}(\psi, \lambda \mid \mathbf{t}). \tag{2.14}$$

In equation (2.14), $\lambda$ is *optimized out* for each value of $\psi$, and this implicitly defines a function $\lambda^*(\psi)$ of optimal $\lambda$ values for each value of $\psi$. For example, given the full parameter for the three-layer problem $\theta = (h_1, h_2, h_3)$, we may consider the hopping rate for the first layer as the interest parameter and the other two hopping rates as nuisance parameters, i.e. $\psi(\theta) = h_1$ and $\lambda(\theta) = (h_2, h_3)$, giving

$$\ell_p(h_1 \mid \mathbf{t}) = \sup_{(h_2, h_3)} \hat{\ell}(h_1, h_2, h_3 \mid \mathbf{t}). \tag{2.15}$$

We implement this optimization using MATLAB's *fmincon* function with bound constraints [40]. For each value of the interest parameter, taken over a sufficiently fine grid, the nuisance parameter is optimized out and the previous optimal value is used as the starting guess for the next optimization problem. Uniformly spaced grids of 40 points, defined on the interval $h_i \in [0.05, 0.5]$ for $i = 1, 2, 3$. Results are plotted in terms of the normalized profile likelihood functions.

The likelihood function is often characterized as representing the information that the data contain about the parameters, and the relative likelihood for different parameter values as indicating the relative evidence for these parameter values [2]. As such, a flat profile is indicative of non-identifiability, therefore a lack of information in the data about a parameter [10]. In general, the degree of curvature is related to the inferential precision [10,13,41]. Likelihood-based confidence intervals can be formed by choosing a threshold-relative profile log-likelihood value, which can be approximately calibrated via the $\chi^2$ distribution (or via simulation). For univariate and bivariate profiles, we use thresholds of $-1.92$ and $-3.00$, respectively, which correspond to approximate 95% confidence regions for sufficiently regular problems [2,42]. The points of intersection can be determined using interpolation.

## (f) Case study 1: two layers

In the first instance, we explore the simplest possible heterogeneous problem, which is a system with just two layers, $m = 2$. In this system, we specify $\hat{x}_1 = 30$, $\hat{x}_2 = L = 70$ with $\theta = (0.2, 0.4)$. We begin considering a modest suite of exit time simulations by releasing $R = 100$ particles at $S = 70$, indicated schematically in figure 3a. Profile likelihoods in figure 3b,c lead to maximum-likelihood estimates (MLE) of $\hat{\theta} = (0.5000, 0.1301)$ for the approximate likelihood and $\hat{\theta} = (0.4878, 0.1327)$ with the exact likelihood. Comparing the shapes of the exact and approximate profiles in figure 3b,c indicates that the approximation is reasonable. However, comparing the MLE with the expected values indicates that these data do not lead to accurate estimates; indeed the true values of $h_1$ and $h_2$ are outside of the 95% confidence intervals with this modest set of simulations.

In the electronic supplementary material, S2, figure S2, we consider the simulated coverage of these intervals. We find that the coverage of the 95% confidence intervals based on the exact likelihood is high (close to 100%, indicating the case shown here is 'unlucky'). By contrast, the simulated coverage of the likelihood-based intervals based on the approximate likelihood is lower (closer to 50%) due to a slight overall right shift in the likelihood function and sharp lower interval bounds. However, the likelihood functions are relatively asymmetric in both cases, with associated confidence intervals typically intersecting with upper bound constraints, indicating one-sided practical identifiability at best under both exact and approximate likelihood analyses. Overall, these features suggest that this experimental design leads to low precision, typically one-sided, interval estimates and can enhance any slight bias in the approximate likelihood. We hence now explore more refined experimental designs where we first consider introducing particles at two locations, and then we consider increasing the total number of particles.

We now examine how the identifiability of the hopping rates changes by altering the experimental design. Results in figure 3e–f show profile likelihoods for a similar problem where we release $R = 50$ particles at $S = 70$ and a further $R = 50$ particles at $S = 30$, so that the data we use consist of 100 stochastic realizations in total, as indicated schematically in figure 3d. Here, the MLE based on the approximate and exact likelihoods are $\hat{\theta} = (0.2094, 0.4650)$ and $\hat{\theta} = (0.2141, 0.3467)$, respectively. There are several interesting observations to make about these results. First, the profile likelihoods based on the exact and approximate likelihood function in figure 3e–f compare well, again suggesting that the approximation is reasonably accurate. Second, if we compare profiles in figure 3e–f with those in figure 3b–c we see that releasing particles at two locations leads to accurate estimates, with a relatively small uncertainty in our estimate of $h_1$, and a larger uncertainty in our estimate of $h_2$, which is one-sided identifiable.

A final set of results in figure 3h–i shows approximate and exact profile likelihoods for the experimental design shown schematically in figure 3g, where we release $R = 500$ particles at $S = 100$ and a further $R = 500$ particles at $S = 30$, giving a total of 1000 stochastic realizations. Again, the exact and approximate profile likelihoods compare well, with $\hat{\theta} = (0.2009, 0.4394)$ for the approximate likelihood, and $\hat{\theta} = (0.2046, 0.4311)$ for the exact likelihood. Overall, comparing the suite of results in figure 3 we see that our ability to estimate the parameters is strongly dependent upon the available data, with the uncertainty in our estimates reducing as the quality and quantity of data increase. In particular, we find that a simple experimental design where we have one release point and one capture point, analogous to Andrews' experiments [21,22] in figure 1b, are insufficient for us to reliably estimate hopping rates in the simple two-layer system. However, our ability to estimate $h_1$ and $h_2$ is improved when we release more particles at different locations, as illustrated in figure 3h–i. Despite these challenges, a key outcome of this case study is that the approximate profile likelihoods compare very well with the exact profile likelihoods. This is a useful finding since the approximate profiles are relatively inexpensive to compute.

## (g) Case study 2: three layers

We now consider a three layer problem with $\theta = (h_1, h_2, h_3)$, which means that in addition to constructing univariate profile likelihoods we can also compute bivariate profile likelihoods.

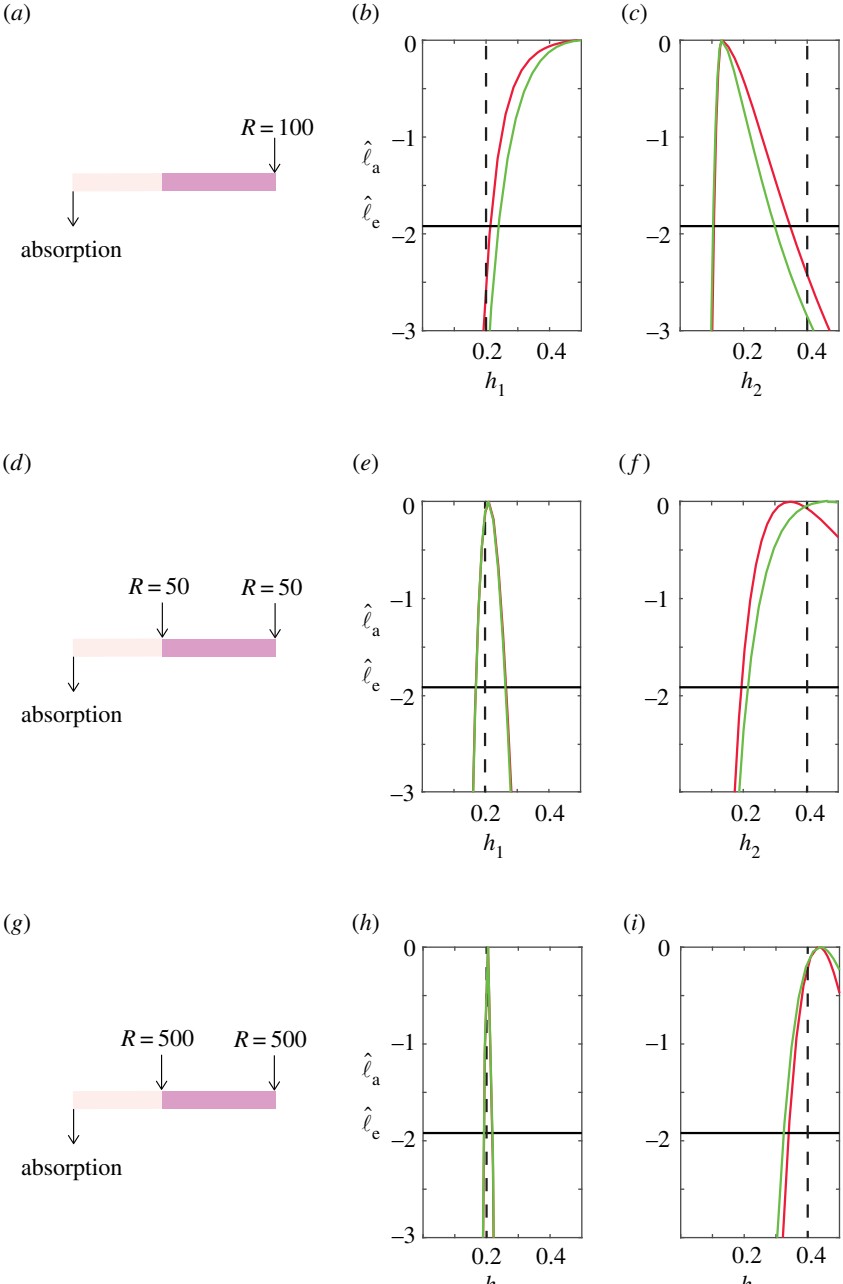

**Figure 3.** Exact and approximate univariate likelihood profiles for a suite of two-layer problems with $\hat{x}_1 = 30$, $\hat{x}_2 = L = 70$ and $\theta = (0.2, 0.4)$. In each row, we show a schematic together with the exact and approximate profile likelihoods in red and green, respectively, with expected value indicated with a vertical dashed black line. (*a–c*) Corresponds to $R = 50$ particles at $S = 100$, giving $\hat{h}_1 = 0.500$ [0.2397, 0.5000] and $\hat{h}_2 = 0.1301$ [0.1036, 0.2975] for the approximate likelihood and $\hat{h}_1 = 0.4878$ [0.2137, 0.5000] and $\hat{h}_2 = 0.1327$ [0.1063, 0.3462] with the exact likelihood. Here, the 95% confidence intervals are given in square brackets, and the $-1.92$ threshold is indicated by the horizontal lines. (*d–f*) Corresponds to $R = 50$ particles at both $S = 30$ and $S = 70$, giving $\hat{h}_1 = 0.2094$ [0.1702, 0.2630] and $\hat{h}_2 = 0.4650$ [0.2155, 0.5000] for the approximate likelihood and $\hat{h}_1 = 0.2141$ [0.1732, 0.2686] and $\hat{h}_2 = 0.3462$ [0.1949, 0.5000] with the exact likelihood. (*g–i*) Corresponds to $R = 500$ particles at both $S = 30$ and $S = 70$, giving $\hat{h}_1 = 0.2009$ [0.1883, 0.2149] and $\hat{h}_2 = 0.4394$ [0.3246, 0.5000] for the approximate likelihood and $\hat{h}_1 = 0.2046$ [0.1922, 0.2177] and $\hat{h}_2 = 0.4311$ [0.3403, 0.5000] using the exact likelihood. (Online version in colour.)

To achieve this, we take the full parameter vector, $\theta = (h_1, h_2, h_3)$, and we consider the hopping rates in the first and second layers as the interest parameters and the hopping rate in the third layer as a nuisance parameter, $\psi(\theta) = (h_1, h_2)$ and $\lambda(\theta) = h_3$. This allows us to evaluate

$$\ell_p(h_1, h_2 \mid \mathbf{t}) = \sup_{h_3} \ell(h_1, h_2, h_3 \mid \mathbf{t}), \tag{2.16}$$

which we compute using MATLAB's *fmincon* function [40]. In this case, we consider a uniform mesh of pairs of the interest parameter and optimize out the nuisance parameter. For our results, we use uniformly spaced grids of $20 \times 20$ points on the interval $h_i \in [0.05, 0.5]$ for $i = 1, 2, 3$.

We now re-visit the same three layer problem from figure 2 with $\hat{x}_1 = 30$, $\hat{x}_2 = 60$ and $\hat{x}_3 = L = 100$, with $\theta = (0.2, 0.3, 0.4)$. Based on our two-layer results in figure 3, here we consider a suite of simulations where we release $R = 50$ particles at each interface location, $S = 30$, $S = 60$ and $S = 100$, giving a total of 150 stochastic simulations as indicated in figure 4a. Using these data, we construct univariate profiles, using both the exact and approximate likelihoods, as shown in figure 4b–d, giving $\hat{\theta} = (0.1882, 0.4636, 0.3856)$ for the approximate likelihood and $\hat{\theta} = (0.1873, 0.3538, 0.3162)$ for the exact likelihood. These results are consistent with the two layer problem since we obtain accurate estimates of $h_1$, but our estimates of $h_2$ and $h_3$ are one-sided identifiable.

We provide additional insight into the identifiability of $\theta$ for the three-layer problem by computing various bivariate profiles. Results in figures 4e–g and 4h–j show the two-dimensional regions that define 95% confidence interval based on the exact (red) and approximate (green) likelihoods, respectively. As in the univariate profiles, these bivariate profiles indicate that we can accurately estimate $h_1$, but $h_2$ and $h_3$ are not identifiable with these data. As in the two-layer problem, one option to improve the identifiability of $h_2$ and $h_3$ would be to collect additional data. For example, one option for this would be to increase the number of points at which particles are released, and another option would be to increase the number of particles released at each point. Varying these aspects of the experimental design is straightforward using the codes provided on GitHub (https://github.com/ProfMJSimpson/StochasticIdentifiability). While generating additional data is straightforward using the simulation model, if we recall the experimental constraints faced by Andrews *et al.* [21], it is clear that simply collecting additional measurements at different spatial resolutions is not always feasible in practice. Therefore, another approach is to interpret the same data using a simpler model, often called *model reduction* [7,8].

## (h) Model reduction

Comparing results in figures 3 and 4 indicates that we can obtain reasonably accurate parameter estimates for a two-layer problem, whereas it can be more challenging to deal with a three-layer problem. Another approach to interpret the three-layer data in figure 4 is to use a simpler, partially homogenized two-layer model. In the partially homogenized model, we consider two layers with $\hat{x}_1 = 30$ and $\hat{x}_2 = L = 100$. This means that we can interpret the first layer, $0 < x < 30$, as being identical in geometry to the first layer in the true three-layer problem, whereas we can interpret the second layer, $30 < x < 100$, as a combined or *effective* layer, with a single transport coefficient, as shown schematically in figure 5a. For this reduced model, we refer to the parameters as $H_1$ and $H_2$. Intuitively, we might expect that $H_1$ would be very close to $h_1$, whereas $H_2$ would be some kind of weighted average of $h_2$ and $h_3$. Using the same data as in figure 4, we construct exact and approximate likelihoods for the reduced model as shown in figure 5b,c. Using the approximate likelihood, we obtain $\hat{\theta} = (0.1887, 0.4411)$, and $\hat{\theta} = (0.1873, 0.3451)$ for the exact likelihood. Comparing results in figure 5b with those in figure 4b, we see that the profiles for $H_1$ and $h_1$ are very similar. In particular, the profile likelihoods for $H_1$ are peaked, the confidence intervals are relatively narrow, and again the exact and approximate profile likelihoods compare very well. If we compare results in figure 5c with those in figure 4c,d we see that the profile for the homogenized hopping rate, $H_2$, is consistent with the hopping rates in the three layer problem, and again the exact and approximate profile likelihoods compare well. This comparison suggests that simplifying the three-layer model into a partially homogenized two-layer model can be a

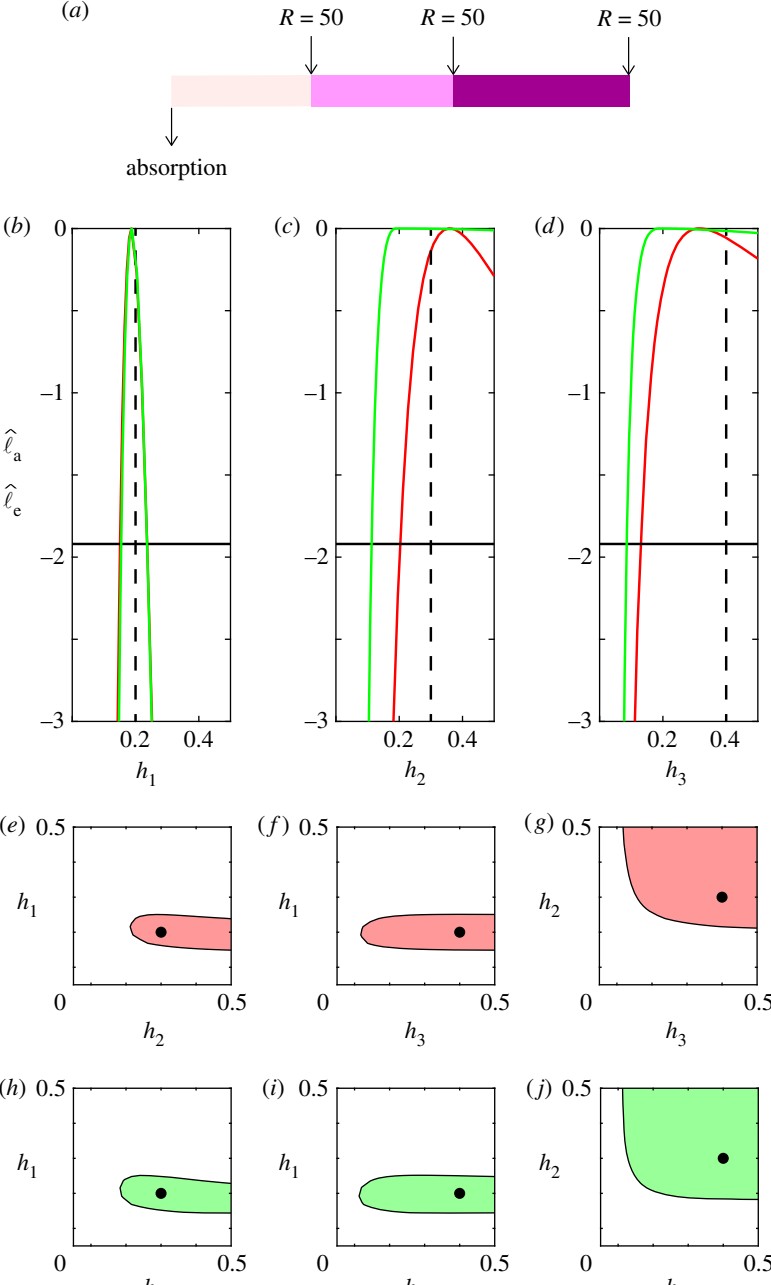

**Figure 4.** Exact and approximate univariate and bivariate profiles for a three-layer problem with $\hat{x}_1 = 30$, $\hat{x}_2 = 30$, $\hat{x}_3 = L = 40$ with $\theta = (0.2, 0.3, 0.4)$, indicated in (*a*). Univariate profile likelihoods for $h_1$, $h_2$ and $h_3$ are shown in (*b*–*d*), respectively, where the exact and approximate results are shown in red and green, respectively, with expected value indicated with a vertical dashed black line. Univariate profiles indicate: $\hat{h}_1 = 0.1882$ [0.1548, 0.2364], $\hat{h}_2 = 0.4636$ [0.1128, 0.5000] and $\hat{h}_3 = 0.3856$ [0.0857, 0.5000] for the approximate likelihood, and $\hat{h}_1 = 0.1873$ [0.1509, 0.2367], $\hat{h}_2 = 0.3538$ [0.2306, 0.5000] and $\hat{h}_3 = 0.3162$ [0.1300, 0.5000] with the exact likelihood. The $-1.92$ threshold is indicated by the horizontal lines. Bivariate profile likelihoods for $(h_1, h_2)$, $(h_1, h_3)$ and $(h_2, h_3)$ pairs are shown in (*e*–*j*). Results in (*e*–*g*) and (*h*–*j*) correspond to the exact and approximate profiles in red and green, respectively. In all bivariate profiles, the 95% confidence region is uniformly shaded and the expected results are shown with a disc. In all cases, we consider $R = 50$ particles released at $S = 30$, $S = 60$ and $S = 100$. (Online version in colour.)

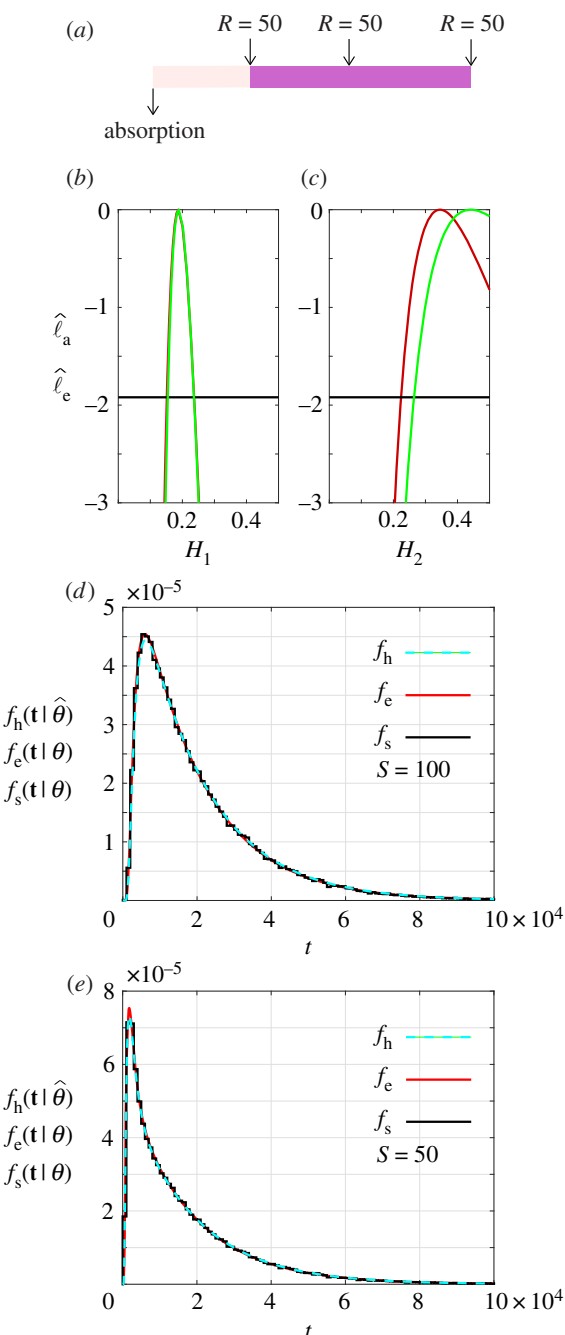

**Figure 5.** Exact and approximate univariate likelihood profiles for a reduced two-layer problem with $\hat{x}_1 = 30$, $\hat{x}_2 = L = 100$ shown schematically in (*a*). Profile likelihoods for $H_1$ and $H_2$ are shown in (*b,c*), where the exact and approximate profile likelihoods are shown in red and green, respectively. These profile likelihoods give MLE values: $\hat{H}_1 = 0.1887$ [0.1546, 0.2360] and $\hat{H}_2 = 0.4411$ [0.2644, 0.500] for the approximate likelihood, and $\hat{H}_1 = 0.1873$ [0.1514, 0.2366] and $\hat{H}_2 = 0.3451$ [0.2240, 0.5000] with the exact likelihood. The $-1.92$ threshold is indicated by the horizontal lines. All results correspond to $R = 50$ particles released at $S = 30$, $S = 60$ and $S = 100$, giving a total of 150 particles. Results in (*d,e*) compare $f_s(\mathbf{t} \mid \theta)$ constructed using $R = 10^4$ three-layer simulations with $\theta = (0.4, 0.3, 0.2)$, with particles released at $S = 100$ and $S = 50$, respectively. In each case, the exit time distribution based on simulation data, $f_s(\mathbf{t} \mid \theta)$, is compared with the exact three-layer distribution, $f_e(\mathbf{t} \mid \theta)$, and the homogenized two-layer distribution calculated using MLE from the exact likelihood, $f_h(\mathbf{t} \mid \hat{\theta})$. Both $f_e(\mathbf{t} \mid \theta)$ and $f_h(\mathbf{t} \mid \hat{\theta})$ are calculated using the exact Markov chain approach outlined in §2d. (Online version in colour.)

useful way of obtaining insight in the face of constraints that prevent us from simply collecting more and more data [21]. We repeated this profiling exercise using additional data (electronic supplementary material, figure S3), showing that the profiles become narrower as the number of particles increases, as expected.

To conclude we compare the ability of the reduced, partially homogenized two-layer model to capture the full three-layer exit time distributions. Results in figure 5*d* show $f_s(\mathbf{t} \mid \theta)$ and $f_e(\mathbf{t} \mid \theta)$ for the full three-layer problem with $\theta = (0.2, 0.3, 0.4)$ and $S = 100$. As expected, this comparison shows that the exact Markov chain result compares very well with simulation data from the three layer problem. We also superimpose the exact distribution for the reduced, partially homogenized two-layer model with $\hat{\theta} = (0.1873, 0.3451)$, which we denote $f_h(\mathbf{t} \mid \hat{\theta})$. At this scale, $f_e(\mathbf{t} \mid \theta)$ is visually indistinguishable from $f_h(\mathbf{t} \mid \hat{\theta})$, confirming that the reduced, partially homogenized model can accurately capture the exit time distribution of the full three-layer problem with $S = 100$. A similar comparison in figure 5*e* with particles released at $S = 50$ further confirms that the reduced, partially homogenized two-layer model can be used to capture the exit time distribution for particles released at different locations. These results, overall, point to model reduction through partial homogenization as a useful strategy to gain partial insight into the spatial structure of particle hopping rates in a heterogeneous material with these kinds of data.

All results and discussion in this work focus on two- and three-layer systems since the original work of Andrews *et al.* [21,22] involves a three-layered material. For completeness, we have also applied our methods to a simple homogeneous single layer and a more complicated four-layer problem (electronic supplementary material, figures S3–S4), where we see that all methodologies and trends established here for the two- and three-layered examples also carry across to four layers.

## 3. Conclusion and outlook

In this work, we analyse parameter identifiability of a stochastic model of diffusive transport. Motivated by the heat transfer experiments reported by Andrews *et al.* [21,22], we consider a one-dimensional model of diffusion through a layered structure that is spatially discretized with a unit lattice. The domain consists of several layers where the hopping rate of particles can be different in each layer. With a reflecting boundary condition at one end of the domain and an absorbing boundary condition at the other, simulations are performed by releasing particles at a particular location, $S$, and the simulation proceeds until the particle is captured at the absorbing boundary and the duration of time required for absorption is recorded. We are interested to explore the following question: give the number of layers, $m$, and the positions of the interfaces, $\hat{x}_0, \hat{x}_1, \ldots, \hat{x}_m$, can we use the exit time data to estimate the hopping rates in the layers, $\theta = (h_1, h_2, \ldots, h_m)$? In particular, we address this question using a profile likelihood analysis, and our approach involves several novel features. Most previous profile likelihood analyses in the literature focus on deterministic process models, such as ordinary differential equations or partial differential equations. These approaches require the separate specification of a process model and a noise model, such as a zero mean, constant variance Gaussian noise. Once the process and noise models are specified, a likelihood function can be formed and numerical optimization methods can be implemented to compute the profile likelihood. Our work is different since we focus on a stochastic process model and this avoids the need for specifying a separate noise model. This can be advantageous since the usual adoption of Gaussian noise models can be inappropriate for count data [23], density data [11] or length data [43], or any other data that are, by definition, non-negative. While the work here is motivated by experimental observations of diffusive transport in the context of heat transfer through a heterogeneous material, there are also many other applications, particularly in biophysics, that involve diffusive transport through layered structures where it is difficult to make observations at high spatial resolution [44,45].

Our approach to compute the likelihood for our model is insightful since we have the opportunity to construct both an exact, albeit computationally expensive likelihood, as well as an approximate but computationally cheap likelihood. The exact likelihood can be obtained by

analysing the random walk model as a Markov chain provides an excellent match with simulation data, whereas the approximate likelihood captures the main features of the model by treating the distribution of exit times as a Gamma function whose first two moments match those obtained by analysing the continuum limit description of the stochastic model. Working with a simple two-layer problem, $\theta = (h_1, h_2)$, we show that the experimental designs used by Andrews *et al.* [21] can lead to inaccurate estimates if the number of realizations is small. Our work provides insight by showing that working with increased data quality and data quantity we can obtain increasingly reliable estimates of the hopping rates, and in all cases, we consider the results obtained using the approximate likelihood are reasonably accurate. Extending the analysis to deal with a three-layer problem, $\theta = (h_1, h_2, h_3)$, again shows that the hopping rate in the first layer, $h_1$, can be estimated reasonably easily, whereas it can be more difficult to estimate the other hopping rates, $h_2$ and $h_3$. One way of dealing with this challenge is to simply incorporate an increasing amount of data. However, since this might not always be possible owing to experimental constraints, we show that another way forward is to interpret the three-layer data using a reduced, partially homogenized two-layer model, for which we can obtain reliable parameter estimates. Indeed, we show that we can accurately capture exit time distributions from the three-layer system using an appropriately parametrized two-layer model.

There are many opportunities to explore extensions of the present study, both in terms of extending the mathematical modelling framework, and in terms of extending the approximations that we invoke for the likelihood function. In terms of the stochastic model, here we consider the most fundamental stochastic model that incorporates an unbiased random walk without particle decay or bias. Further extensions would be to consider a more general model that incorporates biased motion with decay. This would involve specifying a unique decay rate for particles in each layer and a separate bias parameter in each layer [31]. Incorporating such mechanisms could be helpful to describe the loss of heat due to perfusion into the blood supply in Andrews' experiments [21]. However, such an extension would triple the size of the parameter vector. For example, in the three layer problem, $m = 3$, the model would involve specifying three hopping rates, three bias parameters and three decay rates. Although we could follow an analogous procedure to analyse such an extended model in terms of both the exact probability mass function, $f_e(t \mid \theta)$, and an approximate probability mass function, $f_a(t \mid \theta)$, we do not follow this procedure here since the kinds of data we are working with lead to identifiability issues for the simpler, more fundamental case where bias and decay are not incorporated into the model. All work in this study is restricted to random walks in one dimension, and the decision to restrict this work to one dimension is because of the geometry of the heat conduction experiments reported by Andrews *et al.* [21,22]. In principle, all approaches outlined in this work can be applied to higher dimensional problems. For example, Carr *et al.* [30] show how to implement the stochastic random walk model for two- and three-dimensional problems where the hopping rate is piecewise constant [30]. For these higher dimensional problems, $M_1(\mathbf{x})$ and $M_2(\mathbf{x})$ are given by the solution of a set of partial differential equations that do not have closed-form solutions. Therefore, the boundary value problems for $M_1(\mathbf{x})$ and $M_2(\mathbf{x})$ would have to be solved numerically, as demonstrated by Carr *et al.* [29], or approximately using perturbation methods, as demonstrated by Simpson *et al.* [46]. Once the numerical or perturbation estimates of $M_1(\mathbf{x})$ and $M_2(\mathbf{x})$ are obtained, the analysis outlined here for the one-dimensional case would apply without change. Of course, in this first preliminary attempt at identifiability analysis of a stochastic spatial model, we have restricted ourselves to the most fundamental problem of working with a one-dimensional domain. However, in the future it would be very interesting to explore how these ideas apply to higher dimensional problems, and we leave this for future consideration.

Other extensions of this work would involve developing different, perhaps more accurate approximations of the probability mass function, $f_a(t \mid \theta)$. In this work, we approximate $f(t \mid \theta)$ as a Gamma distribution, $\Gamma(t; a, b)$, with $a$ and $b$ chosen so that the first two moments match the first two moments of exit time when the stochastic model is analysed in terms of the continuum limit. The choice of using the Gamma distribution to approximate the exit time distribution is convenient, but not necessary. The approximation is convenient since it

is straightforward to choose $a$ and $b$ to match the first two moments, and our results show that the approximation can be very accurate. However, other approximations, including the generalized Gamma distribution with three free parameters could be used to develop alternative approximations. In this case, the parameters could be chosen to match the first three moments of exit time in the continuum limit analogue of the stochastic model [31]. Additional candidates for developing alternative approximation would be a phase type distribution, where additional moments can be incorporated to refine the approximation.

Returning to the indirect inference interpretation of our approximate method, here we were able to directly compute the mapping between the auxiliary parameter and parameter of interest. However, for a large class of complex stochastic models, it will not be feasible to obtain such a direct mapping. In such cases, it becomes necessary to estimate the mapping via simulation, and then dealing with a noisy approximate likelihood. We plan to explore such models in future research.

Data accessibility. MATLAB implementations of all computations are available on GitHub: https://github.com/ProfMJSimpson/StochasticIdentifiability.

Authors' contributions. All authors conceived the study. M.J.S., A.P.B and R.E.B. conceived the study. M.J.S., A.P.B. and E.J.C. carried out the stochastic simulations, performed the mathematical derivations, and wrote code to implement the algorithms. All authors interpreted the results. M.J.S and A.P.B. drafted the manuscript, and all authors edited the manuscript and approved the final version.

Competing interests. We declare we have no competing interests.

Funding. M.J.S. is supported by the Australian Research Council (DP200100177). C.D. is supported by the Australian Research Council (DP200102101). O.J.M. is supported by the University of Auckland, Faculty of Engineering James and Hazel D. Lord Emerging Faculty Fellowship. R.E.B. is a Royal Society Wolfson Research Merit Award holder and also acknowledges the Biotechnology and Biological Sciences Research Council for funding via grant no. BB/R000816/1.

Acknowledgements. We thank two referees for helpful suggestions.

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
