## [Peer Review File · Proceedings. Mathematical, Physical, and Engineering Sciences]

Review History

RSPA-2020-0902.R0 (Original submission)

Review form: Referee 1

Is the manuscript an original and important contribution to its field?

Good

Is the paper of sufficient general interest?

Excellent

Is the overall quality of the paper suitable?

Good

Can the paper be shortened without overall detriment to the main message?

Yes

Do you think some of the material would be more appropriate as an electronic appendix?

No

Do you have any ethical concerns with this paper?

No

Recommendation?

Major revision is needed (please make suggestions in comments)

Comments to the Author(s)

The authors computed an exact, albeit computationally expensive likelihood, as well as an approximate but computationally cheap likelihood for the model of diffusion in heterogeneous media. The analysis works well in a simple two-layer problem. This work provides insight by showing with increased data quality and data quantity we can obtain increasingly reliable estimates of the hopping rates. However, when the authors extended the analysis to deal with a three-layer problem again, the result showed that the hopping rate in the first layer, h_1 , can be estimated reasonably but it can be more difficult to estimate the other hopping rates, h_2 and h_3 . The authors showed that if the model is reduced to a partially homogenized two-layer model, for which we can obtain reliable parameter estimates.

The results are very clear, and the manuscript is well written. The main weak points of the study are that only a one-dimensional domain was considered and the linear diffusion model they used is too simple. It may not be enough to show the novelty of the work. Although the authors mentioned that this work is motivated by the experimental work of Andrews et al., it seems that the work did not include any case study which is really based on the experimental data from Andrews et al. If more examples based on real experimental data can be provided, the work could show its novelty even though the model is simple.

Some minor comments:

1. Page 5, Line 10, the author can point out that the simulation in Figure 2 is the particular example.
2. Page 14, the model reduction is an important result to extend the analysis to the system with more layers so more examples are necessary for supporting this reduction. Does the method work for a four-layer problem, or more layers.

Review form: Referee 2

Is the manuscript an original and important contribution to its field?

Acceptable

Is the paper of sufficient general interest?

Acceptable

Is the overall quality of the paper suitable?

Marginal

Can the paper be shortened without overall detriment to the main message?

Yes

Do you think some of the material would be more appropriate as an electronic appendix?

No

Do you have any ethical concerns with this paper?

No

Recommendation?

Major revision is needed (please make suggestions in comments)

Comments to the Author(s)

Profile likelihood analysis for a stochastic model of diffusion in heterogeneous media

The authors consider a model for data previously reported by the first author and collaborators concerning the measurement of thermal diffusivity in multiple layers of skin when subjected to high temperatures. Importantly, the only readout from heat being applied at one end is the temperature at the other end. The basic question is whether this experimental data is sufficient to discover whether or not the medium is stratified into multiple heterogeneous layers. My intuition would be that no, that should not be possible, and that seems to be what the authors report as well. They then proceed to propose alternative experimental designs (which they admit are not feasible) and see what an identifiability analysis reveals there.

While I do not disagree with the spirit of this investigation, and I believe that profile likelihoods and the notions of structural vs practical identifiability should appear more in applied math literature, I cannot recommend this work for publication. As detailed below, (1) I have concerns about the basic model proposed; (2, more importantly) I am not sure that profile likelihood analysis was properly executed (or at least properly presented); and (3) the lack of rigorous numerical experimentation makes the model reduction section less compelling.

Regarding concern (1).

(1a) The use of discrete heat particles and first passage times.

The authors are inspired by Ref 24 in which heat is applied at one end of a layer and then measured in a time-dependent way at the other end. They model the problem using discrete particles conducting a random walk along a discrete landscape. In place of discussing the time-dependent readout at the absorbing end, the authors claim it is sufficient to solve (or approximately solve) the first-passage time problem for the individual particles to traverse the entire layer. I believe that I can be convinced of this, but the claim needs more support. (How much does the arrival of a particle at the absorbing end contribute to a rise in temperature? If we increase the applied temperature by 10 degrees, say, what does that do to the increased rate of particle release into the medium?) I ask these questions because the proposed "experiments" involve release specific numbers of particles (50 to 500) in specific locations. In practical terms, does this just mean applying a higher temperature? In the experiments, is the applied heat instantaneous, or is it held there for a moment? If it is held there for any substantial time, then it might be difficult to infer first-passage times for particles across the layer. I worry that there is a significant gap between this analysis and something that can be confronted with real data.

(1b) The use of a discrete-state space in the approximation of first passage times.

Accepting the premise that it is sufficient to study first-passage times of "heat particles," it would be preferable to use Brownian particles diffusing in continuous space. The authors decide instead to discretize both space and time. If this discretization was close to the continuum limit, I would have no issue with this. However, the chosen parameters are far from this limit and I am concerned that this might have a profound impact on their simulated first passage times. To see why, consider the case where the heat parameter h is 0.4. This means, at the end of the time duration τ , that a particle in the region of interest has a 0.4 chance of appearing one step to the right and a 0.4 chance of appearing to the left, with a 0.2 chance of staying in the same spot. I am concerned that such a particle actually has a very high chance of making multiple steps within the time duration, and as a result first passage times could be seriously underestimated. To see why, consider standard Brownian motion (mean zero, diffusivity one). Suppose our discretization of space is 0.1 and the particle starts at location 0. A quick calculation shows that the amount of time necessary to pass for Brownian motion to have a 0.2 chance of staying in an interval of size 0.1, specifically the interval $(-0.05, 0.05)$, is about 0.04. We can then calculate the probability that this Brownian motion will be in the interval $(-0.15, -0.05)$, which is 0.18; the probability it appears in $(-0.25, -0.15)$, which is 0.12; and the probability it appears beyond -0.25 , which is 0.1. The details of my hurried calculation might be off, but this says that for the given parameter choice, there is a 20 percent chance the particle will have moved at least three sites away (left or right) in the given time interval. If the number of sites is small (the authors use 30 at

times), it seems like this difference could add up. And note that several confidence intervals in the later analysis include the extreme case $h=0.5$!

My guess is that the authors want to use the moment calculations (25)-(27) and similar formulas are not available for Brownian motion passing through heterogeneous layers? It would just be reassuring to know that here, or elsewhere, these approximations have been vetted when we are far from the continuum limit.

(2) I would like to zero in on the first example provided in Section 2(f), which is the most in spirit like the experiments that inspired this work. The true parameters are 0.2 and 0.4, but the observed profile likelihood for h_1 has a plateau, implying unidentifiability. In fact, the calculated MLE is 0.5 for h_1 (not displayed in the figure) and 0.13 for h_2 , with similar values for the approximate method.

The authors write:

"However, comparing the MLE with the expected values indicates that this data does not lead to accurate estimates, indeed the true values of h_1 and h_2 are outside of the 95% confidence intervals with this modest set of simulations. Despite the fact that, in this instance, the true parameter values do not lie within the identified interval, different realisations of the same experimental design typically lead to confidence intervals that do contain the true values."

This statement makes it seem like the authors just do not want to spend time on this example, but frankly this is an important place to make sure everything is working as it should. Moreover it is a nice opportunity to walk the reader through unidentifiability in this problem. Is it true that a larger simulation set will lead to confidence intervals that are not much better? How do we even know how many particles are needed? In all honesty, the authors should consider simulating a one-layer problem ($L = 70$, $h=0.25$ or 0.3) and see how many particles are needed to provide a tight confidence interval. Claims of unidentifiability get stronger when we see what it takes to get positive identifiability.

There also seems to be a missed opportunity to understand the relationship between estimates of the two parameters. When h_1 is high, it stands to reason that h_2 must compensate and be low. But the opposite relation does not seem to hold since the inferred values for h_2 are all low? And why is it that h_1 has such a wide range of viable values, but h_2 does not?

(3) I find the model reduction section at the end somewhat unconvincing. The authors claim that not much is lost when replacing a three layer problem with a two layer problem, but this seems to be because there is no information in either case. The confidence interval for H_2 is $[0.224, 0.500]$ (essentially saying it is unidentifiable) as are h_2 and h_3 for the three layer problem. I suspect that if the simulations were greatly increased, at least h_2 would become identifiable and at that point seeing if model reduction is still viable would be meaningful. As of right now, I feel like the statement is "since we don't have enough data to specify the parameters of a complex model, it is better to have a less complex model that we cannot specify."

In spirit I want to like this paper, but the analysis feels incomplete. There seem to be missed opportunities, both in not connecting the model realistically to the data and in not carefully developing the numerical experiments in order to make a clear study of practical and structural identifiability.

Decision letter (RSPA-2020-0902.R0)

16-Feb-2021

Dear Professor Simpson:

I am writing to inform you that your manuscript RSPA-2020-0902 entitled "Profile likelihood analysis for a stochastic model of diffusion in heterogeneous media" has been rejected in its present form for publication in Proceedings A.

The Editor has made this decision based on the advice of referees, and taking into account their own opinion of your paper. With this in mind we would like to invite a resubmission, provided the comments of the referees and any comments from the Editor are taken into account. This is not a provisional acceptance.

The resubmission will be treated as a new manuscript. Please note that resubmissions must be submitted within six months of the date of this email. In exceptional circumstances, extensions may be possible if agreed with the Editorial Office.

Please find below the comments made by the referees, not including confidential reports to the Editor, which I hope you will find useful. If you do choose to resubmit your manuscript, please include details of how you have responded to the comments, and the adjustments you have made.

Please note that we have a strict upper limit of 28 pages for each paper. Please endeavour to incorporate any revisions while keeping the paper within journal limits. Please note that page charges are made on all papers longer than 20 pages. If you cannot pay these charges you must reduce your paper to 20 pages before submitting your revision. Your paper has been ESTIMATED to be 18 pages. We cannot proceed with typesetting your paper without your agreement to meet page charges in full should the paper exceed 20 pages when typeset. If you have any questions, please do get in touch.

To upload a resubmitted manuscript, log into <http://mc.manuscriptcentral.com/prsa> and enter your Author Centre, where you will find your manuscript title listed under "Manuscripts with Decisions." Under "Actions," click on "Create a Resubmission." Please be sure to indicate that it is a resubmission, and ensure you enter this ID - RSPA-2020-0902 - as the previous submission number.

Yours sincerely
Raminder Shergill
proceedingsa@royalsociety.org

on behalf of
Professor Matjaz Perc
Board Member
Proceedings A

Reviewer(s)' Comments to Author:

Referee: 1

Comments to the Author(s)

The authors computed an exact, albeit computationally expensive likelihood, as well as an approximate but computationally cheap likelihood for the model of diffusion in heterogeneous media. The analysis works well in a simple two-layer problem. This work provides insight by showing with increased data quality and data quantity we can obtain increasingly reliable

estimates of the hopping rates. However, when the authors extended the analysis to deal with a three-layer problem again, the result showed that the hopping rate in the first layer, h_1 , can be estimated reasonably but it can be more difficult to estimate the other hopping rates, h_2 and h_3 . The authors showed that if the model is reduced to a partially homogenized two-layer model, for which we can obtain reliable parameter estimates.

The results are very clear, and the manuscript is well written. The main weak points of the study are that only a one-dimensional domain was considered and the linear diffusion model they used is too simple. It may not be enough to show the novelty of the work. Although the authors mentioned that this work is motivated by the experimental work of Andrews et al., it seems that the work did not include any case study which is really based on the experimental data from Andrews et al. If more examples based on real experimental data can be provided, the work could show its novelty even though the model is simple.

Some minor comments:

1. Page 5, Line 10, the author can point out that the simulation in Figure 2 is the particular example.
2. Page 14, the model reduction is an important result to extend the analysis to the system with more layers so more examples are necessary for supporting this reduction. Does the method work for a four-layer problem, or more layers.

Referee: 2

Comments to the Author(s)

Profile likelihood analysis for a stochastic model of diffusion in heterogeneous media

The authors consider a model for data previously reported by the first author and collaborators concerning the measurement of thermal diffusivity in multiple layers of skin when subjected to high temperatures. Importantly, the only readout from heat being applied at one end is the temperature at the other end. The basic question is whether this experimental data is sufficient to discover whether or not the medium is stratified into multiple heterogeneous layers. My intuition would be that no, that should not be possible, and that seems to be what the authors report as well. They then proceed to propose alternative experimental designs (which they admit are not feasible) and see what an identifiability analysis reveals there.

While I do not disagree with the spirit of this investigation, and I believe that profile likelihoods and the notions of structural vs practical identifiability should appear more in applied math literature, I cannot recommend this work for publication. As detailed below, (1) I have concerns about the basic model proposed; (2, more importantly) I am not sure that profile likelihood analysis was properly executed (or at least properly presented); and (3) the lack of rigorous numerical experimentation makes the model reduction section less compelling.

Regarding concern (1).

(1a) The use of discrete heat particles and first passage times.

The authors are inspired by Ref 24 in which heat is applied at one end of a layer and then measured in a time-dependent way at the other end. They model the problem using discrete particles conducting a random walk along a discrete landscape. In place of discussing the time-dependent readout at the absorbing end, the authors claim it is sufficient to solve (or approximately solve) the first-passage time problem for the individual particles to traverse the entire layer. I believe that I can be convinced of this, but the claim needs more support. (How much does the arrival of a particle at the absorbing end contribute to a rise in temperature? If we increase the applied temperature by 10 degrees, say, what does that do to the increased rate of particle release into the medium?) I ask these questions because the proposed "experiments" involve release specific numbers of particles (50 to 500) in specific locations. In practical terms, does this just mean applying a higher temperature? In the experiments, is the applied heat instantaneous, or is it held there for a moment? If it is held there for any substantial time, then it

might be difficult to infer first-passage times for particles across the layer. I worry that there is a significant gap between this analysis and something that can be confronted with real data.

(1b) The use of a discrete-state space in the approximation of first passage times.

Accepting the premise that it is sufficient to study first-passage times of "heat particles," it would be preferable to use Brownian particles diffusing in continuous space. The authors decide instead to discretize both space and time. If this discretization was close to the continuum limit, I would have no issue with this. However, the chosen parameters are far from this limit and I am concerned that this might have a profound impact on their simulated first passage times. To see why, consider the case where the heat parameter h is 0.4. This means, at the end of the time duration τ , that a particle in the region of interest has a 0.4 chance of appearing one step to the right and a 0.4 chance of appearing to the left, with a 0.2 chance of staying in the same spot. I am concerned that such a particle actually has a very high chance of making multiple steps within the time duration, and as a result first passage times could be seriously underestimated. To see why, consider standard Brownian motion (mean zero, diffusivity one). Suppose our discretization of space is 0.1 and the particle starts at location 0. A quick calculation shows that the amount of time necessary to pass for Brownian motion to have a 0.2 chance of staying in an interval of size 0.1, specifically the interval $(-0.05, 0.05)$, is about 0.04. We can then calculate the probability that this Brownian motion will be in the interval $(-0.15, -0.05)$, which is 0.18; the probability it appears in $(-0.25, -0.15)$, which is 0.12; and the probability it appears beyond -0.25 , which is 0.1. The details of my hurried calculation might be off, but this says that for the given parameter choice, there is a 20 percent chance the particle will have moved at least three sites away (left or right) in the given time interval. If the number of sites is small (the authors use 30 at times), it seems like this difference could add up. And note that several confidence intervals in the later analysis include the extreme case $h=0.5$!

My guess is that the authors want to use the moment calculations (25)-(27) and similar formulas are not available for Brownian motion passing through heterogeneous layers? It would just be reassuring to know that here, or elsewhere, these approximations have been vetted when we are far from the continuum limit.

(2) I would like to zero in on the first example provided in Section 2(f), which is the most in spirit like the experiments that inspired this work. The true parameters are 0.2 and 0.4, but the observed profile likelihood for h_1 has a plateau, implying unidentifiability. In fact, the calculated MLE is 0.5 for h_1 (not displayed in the figure) and 0.13 for h_2 , with similar values for the approximate method.

The authors write:

"However, comparing the MLE with the expected values indicates that this data does not lead to accurate estimates, indeed the true values of h_1 and h_2 are outside of the 95% confidence intervals with this modest set of simulations. Despite the fact that, in this instance, the true parameter values do not lie within the identified interval, different realisations of the same experimental design typically lead to confidence intervals that do contain the true values."

This statement makes it seem like the authors just do not want to spend time on this example, but frankly this is an important place to make sure everything is working as it should. Moreover it is a nice opportunity to walk the reader through unidentifiability in this problem. Is it true that a larger simulation set will lead to confidence intervals that are not much better? How do we even know how many particles are needed? In all honesty, the authors should consider simulating a one-layer problem ($L = 70$, $h=0.25$ or 0.3) and see how many particles are needed to provide a tight confidence interval. Claims of unidentifiability get stronger when we see what it takes to get positive identifiability.

There also seems to be a missed opportunity to understand the relationship between estimates of the two parameters. When h_1 is high, it stands to reason that h_2 must compensate and be low.

But the opposite relation does not seem to hold since the inferred values for h_2 are all low? And why is it that h_1 has such a wide range of viable values, but h_2 does not?

(3) I find the model reduction section at the end somewhat unconvincing. The authors claim that not much is lost when replacing a three layer problem with a two layer problem, but this seems to be because there is no information in either case. The confidence interval for H_2 is $[0.224, 0.500]$ (essentially saying it is unidentifiable) as are h_2 and h_3 for the three layer problem. I suspect that if the simulations were greatly increased, at least h_2 would become identifiable and at that point seeing if model reduction is still viable would be meaningful. As of right now, I feel like the statement is "since we don't have enough data to specify the parameters of a complex model, it is better to have a less complex model that we cannot specify."

In spirit I want to like this paper, but the analysis feels incomplete. There seem to be missed opportunities, both in not connecting the model realistically to the data and in not carefully developing the numerical experiments in order to make a clear study of practical and structural identifiability.

Author's Response to Decision Letter for (RSPA-2020-0902.R0)

See Appendix A.

RSPA-2021-0214.R0

Review form: Referee 1

Is the manuscript an original and important contribution to its field?

Good

Is the paper of sufficient general interest?

Excellent

Is the overall quality of the paper suitable?

Excellent

Can the paper be shortened without overall detriment to the main message?

Yes

Do you think some of the material would be more appropriate as an electronic appendix?

No

Do you have any ethical concerns with this paper?

No

Recommendation?

Accept as is

Comments to the Author(s)

The responses from the authors are fine.

Review form: Referee 2**Is the manuscript an original and important contribution to its field?**

Acceptable

Is the paper of sufficient general interest?

Good

Is the overall quality of the paper suitable?

Good

Can the paper be shortened without overall detriment to the main message?

Yes

Do you think some of the material would be more appropriate as an electronic appendix?

No

Do you have any ethical concerns with this paper?

No

Recommendation?

Accept with minor revision (please list in comments)

Comments to the Author(s)

The authors have made substantial edits to the paper in response to both the other reviewer and myself. Their response was thorough and, while I still have misgivings about the use of a discretized model in calculating the first-passage time distribution, I do not believe that concern should be a barrier to publication. The new text in the introduction, in particular, does a very nice job of motivating the context and the mode of investigation. Moreover, the new material included in the supplemental information provides further support for their modeling decisions and does inspire confidence that the inference method converges to the correct parameter values (when possible).

I would just like to make two notes: neither of which necessitate changes in the text, but I would like to share them with the authors.

First, regarding the use of the discrete model. I certainly understand the convergence of the discrete model to the continuum model in the $\tau \rightarrow 0$ limit. My concern is that the parameter range involves values of h that seem to imply that we are far from the $\tau \rightarrow 0$ limit. Generally, when I think about a random walk model approximating particle based solutions to the heat equation, I am thinking about the random walk being a discretization of a Brownian motion (which can take several steps in a single interval of time). It is only in the $\tau \rightarrow 0$ limit that it becomes unlikely that the Brownian motion would only take one step in the time interval. This is important when using random walks to approximate first-passage time distributions for Brownian motion. By definition, the fastest walkers will be slower than the fastest Brownian motions because they can only take one step at a time

In Figure 2 and 3, the authors provide a plot that intended to show how faithfully simulated f_{pt} properties are captured by the theory. But this is a self-referential consistency between the discrete model and its own estimator. It is still not clear how these compare to the first-passage

times of Brownian motions with different diffusivities in different layers. Regardless, the authors do point out that their intent is to say something about whether a FPT method would yield identifiable estimation of parameters; and this investigation can be carried out and be interesting and publishable without fully connecting to the continuum model.

Second, the thing that still stands out to me as strange is the way that the method misidentifies the parameters when there are two layers. In the displayed profile likelihoods in Figure 4, the authors share profile likelihoods that exclude the true values from the presented 95% confidence intervals. The authors call this "unlucky", at least for the "exact" likelihood profile, which usually includes the true parameter in the 95% range. But looking at Figure S1, the figure 4 plot looks typical for the approximate likelihood profile. Not only are the true values typically excluded from the confidence interval, their values seem to be typically reversed, with h_2 being estimated to have lower values than h_1 . Why would the exact likelihood have modes that are consistent with the real values, but the approximate likelihood have modes that are reversed?

To their credit, the authors do not hide any of this. With the new information in the SI, the reader has enough information to understand what is being done and see when the method is and is not successful, perhaps inspiring other studies.

Decision letter (RSPA-2021-0214.R0)

10-May-2021

Dear Dr Simpson,

On behalf of the Editor, I am pleased to inform you that your Manuscript RSPA-2021-0214 entitled "Profile likelihood analysis for a stochastic model of diffusion in heterogeneous media" has been accepted for publication subject to minor revisions in Proceedings A. Please find the referees' comments below.

The reviewer(s) have recommended publication, but also suggest some minor revisions to your manuscript. Therefore, I invite you to respond to the reviewer(s)' comments and revise your manuscript. Please note that we have a strict upper limit of 28 pages for each paper. Please endeavour to incorporate any revisions while keeping the paper within journal limits. Please note that page charges are made on all papers longer than 20 pages. If you cannot pay these charges you must reduce your paper to 20 pages before submitting your revision. Your paper has been ESTIMATED to be 21 pages. We cannot proceed with typesetting your paper without your agreement to meet page charges in full should the paper exceed 20 pages when typeset. If you have any questions, please do get in touch.

It is a condition of publication that you submit the revised version of your manuscript within 7 days. If you do not think you will be able to meet this date please let me know in advance of the due date.

To revise your manuscript, log into <https://mc.manuscriptcentral.com/prsa> and enter your Author Centre, where you will find your manuscript title listed under "Manuscripts with Decisions." Under "Actions," click on "Create a Revision." Your manuscript number has been appended to denote a revision.

You will be unable to make your revisions on the originally submitted version of the manuscript. Instead, revise your manuscript and upload a new version through your Author Centre.

When submitting your revised manuscript, you will be able to respond to the comments made by the referee(s) and upload a file "Response to Referees" in Step 1: "View and Respond to Decision Letter". You can use this to document any changes you make to the original manuscript. In order to expedite the processing of the revised manuscript, please be as specific as possible in your response to the referee(s).

IMPORTANT: Your original files are available to you when you upload your revised manuscript. Please delete any redundant files before completing the submission process.

When uploading your revised files, please make sure that you include the following as we cannot proceed without these:

- 1) A text file of the manuscript (doc, txt, rtf or tex), including the references, tables (including captions) and figure captions. Please remove any tracked changes from the text before submission. PDF files are not an accepted format for the "Main Document".
- 2) A separate electronic file of each figure (tif, eps or print-quality pdf preferred). The format should be produced directly from original creation package, or original software format.
- 3) Electronic Supplementary Material (ESM): all supplementary materials accompanying an accepted article will be treated as in their final form. Note that the Royal Society will not edit or typeset supplementary material and it will be hosted as provided. Please ensure that the supplementary material includes the paper details where possible (authors, article title, journal name). Supplementary files will be published alongside the paper on the journal website and posted on the online figshare repository (<https://figshare.com>). The heading and legend provided for each supplementary file during the submission process will be used to create the figshare page, so please ensure these are accurate and informative so that your files can be found in searches. Files on figshare will be made available approximately one week before the accompanying article so that the supplementary material can be attributed a unique DOI.

Alternatively you may upload a zip folder containing all source files for your manuscript as described above with a PDF as your "Main Document". This should be the full paper as it appears when compiled from the individual files supplied in the zip folder.

Article Funder

Please ensure you fill in the Article Funder question on page 2 to ensure the correct data is collected for FundRef (<http://www.crossref.org/fundref/>).

Media summary

Please ensure you include a short non-technical summary (up to 100 words) of the key findings/importance of your paper. This will be used for to promote your work and marketing purposes (e.g. press releases). The summary should be prepared using the following guidelines:

- *Write simple English: this is intended for the general public. Please explain any essential technical terms in a short and simple manner.
- *Describe (a) the study (b) its key findings and (c) its implications.
- *State why this work is newsworthy, be concise and do not overstate (true 'breakthroughs' are a rarity).
- *Ensure that you include valid contact details for the lead author (institutional address, email address, telephone number).

Cover images

We welcome submissions of images for possible use on the cover of Proceedings A. Images should be square in dimension and please ensure that you obtain all relevant copyright permissions before submitting the image to us. If you would like to submit an image for consideration please send your image to proceedingsa@royalsociety.org

Open Access

You are invited to opt for open access, our author pays publishing model. Payment of open access fees will enable your article to be made freely available via the Royal Society website as soon as it is ready for publication. For more information about open access please visit <https://royalsociety.org/journals/authors/open-access/>. The open access fee for this journal is £1700/\$2380/€2040 per article. VAT will be charged where applicable. Please note that if the corresponding author is at an institution that is part of a Read and Publishing deal you are required to select this option. See <https://royalsociety.org/journals/librarians/purchasing/read-and-publish/read-publish-agreements/> for further details.

Once again, thank you for submitting your manuscript to Proceedings A and I look forward to receiving your revision. If you have any questions at all, please do not hesitate to get in touch.

Best wishes
Raminder Shergill
proceedingsa@royalsociety.org
Proceedings A

on behalf of
Professor Matjaz Perc
Board Member
Proceedings A

Reviewer(s)' Comments to Author:

Referee: 1

Comments to the Author(s)
The responses from the authors are fine.

Referee: 2

Comments to the Author(s)
The authors have made substantial edits to the paper in response to both the other reviewer and myself. Their response was thorough and, while I still have misgivings about the use of a discretized model in calculating the first-passage time distribution, I do not believe that concern should be a barrier to publication. The new text in the introduction, in particular, does a very nice job of motivating the context and the mode of investigation. Moreover, the new material included in the supplemental information provides further support for their modeling decisions and does inspire confidence that the inference method converges to the correct parameter values (when possible).

I would just like to make two notes: neither of which necessitate changes in the text, but I would like to share them with the authors.

First, regarding the use of the discrete model. I certainly understand the convergence of the discrete model to the continuum model in the $\tau \rightarrow 0$ limit. My concern is that the parameter range involves values of h that seem to imply that we are far from the $\tau \rightarrow 0$ limit. Generally, when I think about a random walk model approximating particle based solutions to the heat equation, I am thinking about the random walk being a discretization of a Brownian

motion (which can take several steps in a single interval of time). It is only in the $\tau \rightarrow 0$ limit that it becomes unlikely that the Brownian motion would only take one step in the time interval. This is important when using random walks to approximate first-passage time distributions for Brownian motion. By definition, the fastest walkers will be slower than the fastest Brownian motions because they can only take one step at a time

In Figure 2 and 3, the authors provide a plot that intended to show how faithfully simulated FPT properties are captured by the theory. But this is a self-referential consistency between the discrete model and its own estimator. It is still not clear how these compare to the first-passage times of Brownian motions with different diffusivities in different layers. Regardless, the authors do point out that their intent is to say something about whether a FPT method would yield identifiable estimation of parameters; and this investigation can be carried out and be interesting and publishable without fully connecting to the continuum model.

Second, the thing that still stands out to me as strange is the way that the method misidentifies the parameters when there are two layers. In the displayed profile likelihoods in Figure 4, the authors share profile likelihoods that exclude the true values from the presented 95% confidence intervals. The authors call this "unlucky", at least for the "exact" likelihood profile, which usually includes the true parameter in the 95% range. But looking at Figure S1, the figure 4 plot looks typical for the approximate likelihood profile. Not only are the true values typically excluded from the confidence interval, their values seem to be typically reversed, with h_2 being estimated to have lower values than h_1 . Why would the exact likelihood have modes that are consistent with the real values, but the approximate likelihood have modes that are reversed?

To their credit, the authors do not hide any of this. With the new information in the SI, the reader has enough information to understand what is being done and see when the method is and is not successful, perhaps inspiring other studies.

Author's Response to Decision Letter for (RSPA-2021-0214.R0)

See Appendix B.

Decision letter (RSPA-2021-0214.R1)

13-May-2021

Dear Dr Simpson

I am pleased to inform you that your manuscript entitled "Profile likelihood analysis for a stochastic model of diffusion in heterogeneous media" has been accepted in its final form for publication in Proceedings A.

Our Production Office will be in contact with you in due course. You can expect to receive a proof of your article soon. Please contact the office to let us know if you are likely to be away from e-mail in the near future. If you do not notify us and comments are not received within 5 days of sending the proof, we may publish the paper as it stands.

As a reminder, you have provided the following 'Data accessibility statement' (if applicable). Please remember to make any data sets live prior to publication, and update any links as needed when you receive a proof to check. It is good practice to also add data sets to your reference list.

Statement (if applicable): All data and software associated with this work is available on GitHub at <https://github.com/ProfMJSimpson/StochasticIdentifiability>

Open access

You are invited to opt for open access, our author pays publishing model. Payment of open access fees will enable your article to be made freely available via the Royal Society website as soon as it is ready for publication. For more information about open access please visit <https://royalsociety.org/journals/authors/which-journal/open-access/>. The open access fee for this journal is £1700/\$2380/€2040 per article. VAT will be charged where applicable.

Note that if you have opted for open access then payment will be required before the article is published – payment instructions will follow shortly.

If you wish to opt for open access then please inform the editorial office (proceedingsa@royalsociety.org) as soon as possible.

Your article has been estimated as being 20 pages long. Our Production Office will inform you of the exact length at the proof stage.

Proceedings A levies charges for articles which exceed 20 printed pages. (based upon approximately 540 words or 2 figures per page). Articles exceeding this limit will incur page charges of £150 per page or part page, plus VAT (where applicable).

Under the terms of our licence to publish you may post the author generated postprint (ie. your accepted version not the final typeset version) of your manuscript at any time and this can be made freely available. Postprints can be deposited on a personal or institutional website, or a recognised server/repository. Please note however, that the reporting of postprints is subject to a media embargo, and that the status the manuscript should be made clear. Upon publication of the definitive version on the publisher's site, full details and a link should be added.

You can cite the article in advance of publication using its DOI. The DOI will take the form: 10.1098/rspa.XXXX.YYYY, where XXXX and YYYY are the last 8 digits of your manuscript number (eg. if your manuscript number is RSPA-2017-1234 the DOI would be 10.1098/rspa.2017.1234).

For tips on promoting your accepted paper see our blog post: <https://royalsociety.org/blog/2020/07/promoting-your-latest-paper-and-tracking-your-results/>

On behalf of the Editor of Proceedings A, we look forward to your continued contributions to the Journal.

Sincerely,
Raminder Shergill
proceedingsa@royalsociety.org

Date March 9, 2021
Contact Matthew J. Simpson
Phone +61 4 1369 6607
E-mail matthew.simpson@qut.edu.au
Subject Manuscript Resubmission

Appendix A

Professor Matthew J. Simpson

Professor Matjaz Perč
Editorial Board Member,
Proceedings A

School of Mathematical Sciences
Science and Engineering Faculty
Queensland University of Technology
GPO Box 2434, GP Campus
Brisbane, Queensland 4001 Australia

Dear Professor Perč,

I am writing to submit a revised version of our manuscript, "Profile likelihood analysis for a stochastic model of diffusion in heterogeneous media", that has been refereed for publication in *Proceedings A*. We appreciate the helpful comments from both referees, and we have revised the manuscript to address all comments raised. In summary, to address the comments of Referee 1 we have re-written the introduction of the manuscript to include additional data from the original experiments that motivated the work, we extended the results to deal with four layers, and explain how to apply the methodologies to higher dimensional problems. To address the comments of Referee 2 we provide new comparisons between stochastic simulation data and mean-field predictions that confirm that the lattice-based random walk model is very well approximated by the continuum limit moment equations. Further, we provide additional data relating to issues of identifiability in the very simple two-layer experiments with one release point, as well as providing additional results for the very simplest single layer problem in the online Supplementary Material document.

All changes to the manuscript are highlighted in blue font, and a point-by-point response to each comment is attached.

We thank you for your time, and hope that you find the revised manuscript suitable for publication.

Yours sincerely,

Matthew J. Simpson

Referee 1

1. The authors computed an exact, albeit computationally expensive likelihood, as well as an approximate but computationally cheap likelihood for the model of diffusion in heterogeneous media. The analysis works well in a simple two-layer problem. This work provides insight by showing with increased data quality and data quantity we can obtain increasingly reliable estimates of the hopping rates. However, when the authors extended the analysis to deal with a three-layer problem again, the result showed that the hopping rate in the first layer, h_1 , can be estimated reasonably but it can be more difficult to estimate the other hopping rates, h_2 and h_3 . The authors showed that if the model is reduced to a partially homogenized two-layer model, for which we can obtain reliable parameter estimates.

The results are very clear, and the manuscript is well written. The main weak points of the study are that only a one-dimensional domain was considered and the linear diffusion model they used is too simple. It may not be enough to show the novelty of the work. Although the authors mentioned that this work is motivated by the experimental work of Andrews et al., it seems that the work did not include any case study which is really based on the experimental data from Andrews et al. If more examples based on real experimental data can be provided, the work could show its novelty even though the model is simple.

Response: We thank Referee 1 for their positive report, and we agree with the broad description of the study. In the revised manuscript we explicitly include additional motivating data (Figure 1(b)) to show the format of the data reported by Andrews et al. (2016), and to point out that one of the key things that Andrews et al. (2016) measure is the duration of time taken for the thermal disturbance at the surface of the skin to influence the subdermal temperature. We feel that presenting this data helps to motivate our work since it provides clear evidence that a first passage time framework is appropriate.

In the revised manuscript we also briefly discuss other medical and scientific applications that involve diffusion across multilayered material where it is natural to measure time series data rather than full spatiotemporal data. Such applications include chemical diffusion of drugs across layered skin (e.g. Hansen et al. (2013)) and chemical diffusion (or advection diffusion) of contaminants across environmental barriers in landfill design (e.g. Foote (2002); Chen et al. (2015)). In both examples it is either very difficult, or technically impossible, to measure the spatial distribution of dissolved solutes through the layered material, and so it is natural to consider time-series data in the form of a breakthrough curve in the same way that Andrews et al. (2016) consider time series data of temperature at the base of the skin in their experiments. These additional applications are now mentioned in the revised manuscript (Page 3, Paragraph 1).

In the revised manuscript we also point out that the aim of our work is not to deal with one particular set of measurements from a particular experiment, but instead we seek to address the question of whether it is possible to use time-series data obtained from a stochastic model of diffusive transport in a layered media to estimate the diffusivities in each layer (Pages 3–4).

We respectfully disagree with the suggestion that “the one-dimensional linear diffusion model they used is too simple”. This comment might be taken to mean that a more complicated model ought to be used, such as a model that incorporates nonlinear diffusion or a drift-diffusion model. However, since we show that it is very challenging to estimate parameters for the simplest case of multilayer linear diffusion in one-dimension, we strongly feel that it is premature to attempt working with a more complicated model at

this stage. Our modelling philosophy is always to work with the simplest possible model, and to fully understand the limits and capabilities of such a simple model in the first instance before attempting to work with a more complicated model. We agree that, if additional data were available, then a more complicated model might be warranted, and we discuss the possibilities of working with more complicated models or more complicated geometries in the revised Conclusion and Outlook section (Page 17, Paragraph 3). The revised Conclusion and Outlook section also provides an explicit discussion about how the techniques used in the work are applied to problems in higher dimensions (Page 18, Paragraph 1), but again we feel it is reasonable to start working in one dimension since this is the first time that a stochastic model of diffusion in heterogeneous media has been used in this way.

2. Page 5, Line 10, the author can point out that the simulation in Figure 2 is the particular example.

Response: We now mention (Page 7, Paragraph 2) that the comparison of $f_e(t | \theta)$, $f_a(t | \theta)$ and $f_s(t | \theta)$ in Figure 2 is for one particular choice of m and θ .

3. Page 14, the model reduction is an important result to extend the analysis to the system with more layers so more examples are necessary for supporting this reduction. Does the method work for a four-layer problem, or more layers.

Response: We agree that model reduction is important, and we now include additional results in the online Supplementary Material document to show the profiling approach applied to four layers (Section S4, Figure S4-Figure S5). Of course, the approach does apply to more than five layers by following the exact same approach outlined in the study.

Overall, in the revised manuscript we have now presented data for $m = 1, 2, 3$ and 4 layers. Further, we have made our codes freely available on GitHub so that the reader can explore questions of working with other problems with different m and θ separately, as required.

Referee 2

1. The authors consider a model for data previously reported by the first author and collaborators concerning the measurement of thermal diffusivity in multiple layers of skin when subjected to high temperatures. Importantly, the only readout from heat being applied at one end is the temperature at the other end. The basic question is whether this experimental data is sufficient to discover whether or not the medium is stratified into multiple heterogeneous layers. My intuition would be that no, that should not be possible, and that seems to be what the authors report as well. They then proceed to propose alternative experimental designs (which they admit are not feasible) and see what an identifiability analysis reveals there.

While I do not disagree with the spirit of this investigation, and I believe that profile likelihoods and the notions of structural vs practical identifiability should appear more in applied math literature, I cannot recommend this work for publication. As detailed below, (1) I have concerns about the basic model proposed; (2, more importantly) I am not sure that profile likelihood analysis was properly executed (or at least properly presented); and (3) the lack of rigorous numerical experimentation makes the model reduction section less compelling.

Response: We thank Referee 2 for their report, and are glad to know that they agree that profile likelihoods and notions of identifiability ought to appear more in the applied mathematics literature. Referee 2 asks a number of good questions that we will now deal with.

2. (1a) The use of discrete heat particles and first passage times. The authors are inspired by Ref 24 in which heat is applied at one end of a layer and then measured in a time-dependent way at the other end. They model the problem using discrete particles conducting a random walk along a discrete landscape. In place of discussing the time-dependent readout at the absorbing end, the authors claim it is sufficient to solve (or approximately solve) the first-passage time problem for the individual particles to traverse the entire layer. I believe that I can be convinced of this, but the claim needs more support. (How much does the arrival of a particle at the absorbing end contribute to a rise in temperature? If we increase the applied temperature by 10 degrees, say, what does that do to the increased rate of particle release into the medium?) I ask these questions because the proposed “experiments” involve release specific numbers of particles (50 to 500) in specific locations. In practical terms, does this just mean applying a higher temperature? In the experiments, is the applied heat instantaneous, or is it held there for a moment? If it is held there for any substantial time, then it might be difficult to infer first-passage times for particles across the layer. I worry that there is a significant gap between this analysis and something that can be confronted with real data.

Response: Referee 2 is correct to say that Andrews et al. (2016) apply a heat source at the top of the layered skin and then measure the response in temperature at the bottom of the layered structure. To make this clear we have now included a typical data set from Andrews et al. (2016) in Figure 1(b) to explicitly show this. We believe it is useful to show this kind of data because it is clear that we observe a finite duration of time for the thermal disturbance at the surface to be measured at the bottom of the layered structure and so it is natural to consider this problem in a first passage framework in a one-dimensional multilayer domain.

Referee 2 asks a good question about the relationship between the macroscopic thermal conduction problem and the discrete model, in particular, they ask how does the arrival of a particle at the absorbing end contribute to a rise in temperature. If we have a standard one-dimensional macroscopic linear heat conduction problem in a homogeneous material,

$$\frac{\partial U(x, t)}{\partial t} = D \frac{\partial^2 U(x, t)}{\partial x^2}, \quad (1)$$

where $U(x, t)$ is the temperature, it is well-known that this model is related to a microscopic lattice-based random walk where a particle undergoes a position-jump process on a lattice with spacing Δ such that the probability of attempting to undergo a nearest neighbour jump within a time step of duration τ is $P \in [0, 1]$. Standard arguments show that the diffusivity is given by

$$D = \lim_{\Delta \rightarrow 0, \tau \rightarrow 0} \left(\frac{P\Delta^2}{2\tau} \right), \quad (2)$$

where limit is constrained such that the ratio Δ^2/τ remains finite as $\Delta \rightarrow 0$ and $\tau \rightarrow 0$. The derivation linking the discrete and continuum models can be found in standard references (e.g. Hughes 1995; Redner 2009; Codling et al. 2008) that are cited in the main document. Here in this letter, for the sake of brevity, we discuss a single layer problem, but our work (Carr and Simpson, 2019) shows how these ideas directly generalise multilayer diffusion with an arbitrary number of layers.

One way of interpreting such a stochastic simulation is that the temperature, $U(x, t)$, is proportional to the probability of occupancy of the i th lattice site at the j th time step, giving $U(x, t) \propto \mathbb{P}(i\Delta, j\tau)$. This means there is no precise correspondence between particle occupancy and temperature. For example, if we use Equation (1) to model an experiment where the surface of the skin is held at 50°C we would impose a boundary condition $U(0, t) = 50^\circ\text{C}$, whereas if we were to model a second, separate experiment where the surface of the skin is held at 60°C we would impose a different boundary condition, $U(0, t) = 60^\circ\text{C}$. While these two particular problems are different owing to the different boundary conditions, if we non-dimensionalise the governing equation using the imposed boundary temperature, giving a dimensionless temperature $U^*(x, t) = U(x, t)/U(0, t)$, it becomes clear that these two problems are mathematically equivalent. Since the random walk model is presented in a non-dimensional format (Page 7, Final Paragraph), our discrete model naturally accounts for such differences within our framework.

Another related question is about the differences in the initial temperature in Andrew's experiments and our stochastic simulations. Figure 1(b) shows that the initial temperature in the experiments is approximately uniform, $U(x, 0) = 35.2^\circ\text{C}$ whereas the initial occupancy of sites in the stochastic model is zero. These differences can be reconciled simply by shifting the independent variable in Equation (1), $\bar{U}(x, t) = U(x, t) - 35.2$, giving the initial condition $\bar{U}(x, 0) = 0$ in terms of the shifted variable. This initial condition is identical to the initial condition in our stochastic simulations.

Referee 2 asks a good question about the differences in boundary conditions between the experiments reported by Andrews and the stochastic simulations. As we have pointed out in the revised manuscript (Page 3, Final Paragraph) there are differences in the initial condition and boundary conditions in the real experiments and in our simulations. Referee 2 is correct to say that Andrews' experiments involve placing a source of heat at the surface of the skin for a particular duration of time, whereas our study works with discrete experiments where we release one particle at the surface of the domain which is equivalent to the initial condition acting like a delta function in time at the release point. However, capturing these precise details is not the aim of the study. To re-iterate, we seek to address whether time-series measurements at one location for diffusion across layered structures is sufficient to estimate the diffusivities in each layer. This is an important question that is relevant to the experiments reported by Andrews et al. (2016), as well as being relevant to other scientific and technological applications, such as chemical diffusion of drugs over layered biological skin and diffusion of contaminants across complicated layered geotextiles (Page 2, Final Paragraph). This particular question can be addressed regardless of the choice of boundary conditions and initial conditions, and here we choose to work with the simplest possible initial condition (i.e. a single agent placed at a particular location) and the simplest possible boundary conditions (i.e. one reflecting boundary condition and one absorbing boundary condition). So, while we agree that there are differences in boundary condition and initial conditions, the most important feature of the experiments is retained, and that feature is that we use time-series data at one location in space with the aim of trying to understand if this data is sufficient to estimate diffusivities.

3. (1b) The use of a discrete-state space in the approximation of first passage times. Accepting the premise that it is sufficient to study first-passage times of "heat particles" it would be preferable to use Brownian particles diffusing in continuous space. The authors decide instead to discretize both space and time. If this discretization was close to the continuum limit, I would have no issue with this. However, the chosen parameters are far from this limit and I am concerned that this might have a profound impact

on their simulated first passage times. To see why, consider the case where the heat parameter h is 0.4. This means, at the end of the time duration τ , that a particle in the region of interest has a 0.4 chance of appearing one step to the right and a 0.4 chance of appearing to the left, with a 0.2 chance of staying in the same spot. I am concerned that such a particle actually has a very high chance of making multiple steps within the time duration, and as a result first passage times could be seriously underestimated. To see why, consider standard Brownian motion (mean zero, diffusivity one). Suppose our discretization of space is 0.1 and the particle starts at location 0. A quick calculation shows that the amount of time necessary to pass for Brownian motion to have a 0.2 chance of staying in an interval of size 0.1, specifically the interval $(-0.05, 0.05)$, is about 0.04. We can then calculate the probability that this Brownian motion will be in the interval $(-0.15, -0.05)$, which is 0.18; the probability it appears in $(-0.25, -0.15)$, which is 0.12; and the probability it appears beyond -0.25 , which is 0.1. The details of my hurried calculation might be off, but this says that for the given parameter choice, there is a 20 percent chance the particle will have moved at least three sites away (left or right) in the given time interval. If the number of sites is small (the authors use 30 at times), it seems like this difference could add up. And note that several confidence intervals in the later analysis include the extreme case $h=0.5$! My guess is that the authors want to use the moment calculations (25)-(27) and similar formulas are not available for Brownian motion passing through heterogeneous layers? It would just be reassuring to know that here, or elsewhere, these approximations have been vetted when we are far from the continuum limit.

Response: This is a good question and Referee 2 is right to ask whether the continuum-limit approximations for the lattice-based random walk model are accurate. In the original submission we included results in the online Supplementary Material that directly addressed this question by showing that the mean-field expressions for $M_1(x)$ and $M_2(x)$ match very well to appropriately averaged data from the discrete model for a particular three layer problem. We agree that this information is very important since our approximate profile likelihood methods rely on the accuracy of these mean-field expressions. To make this clearer we have now expanded the continuum-discrete comparison and moved this section from the online Supplementary Material document into the main document. In the revised Section 2(c) we compare $M_1(x)$ and $M_2(x)$ with averaged simulation data for three different problems: a two-layer problem, a three-layer problem and a four-layer problem (Section 2c, Pages 8–9, Figure 3). Originally we decided that it would be sufficient to provide this discrete-continuum comparison in the online Supplementary Material document. However, given this question from Referee 2 we think it is reasonable to expand the discrete-continuum comparison and move it into the main document.

Referee 2 asks whether a particle will take multiple steps within the same time step. Note that, in our implementation of the stochastic model this never happens because we use a constant time stepping procedure where it is not possible for a particle to undergo more than one event per step. This is consistent with standard arguments leading to continuum limit descriptions since we take a limit, $\tau \rightarrow 0$, such that the probability of more than one event occurring within a single time step vanishes. The fact that averaged data from our discrete simulations matches the solution of the continuum-limit equations (Section 2c, Pages 8–9, Figure 3) confirms that our discrete model is consistent with the continuum limit description. The full details of our continuum limit derivation have been laid out, in detail, in our previous work (Carr and Simpson, 2019), and other references provide a full discussion of the assumptions implicit in the continuum limit derivation (see Section 2.8 in Codling et al. 2008).

4. (2) I would like to zero in on the first example provided in Section 2(f), which is the most in spirit like the experiments that inspired this work. The true parameters are 0.2 and 0.4, but the observed profile likelihood for h_1 has a plateau, implying unidentifiability. In fact, the calculated MLE is 0.5 for h_1 (not displayed in the figure) and 0.13 for h_2 , with similar values for the approximate method.

The authors write: “However, comparing the MLE with the expected values indicates that this data does not lead to accurate estimates, indeed the true values of h_1 and h_2 are outside of the 95% confidence intervals with this modest set of simulations. Despite the fact that, in this instance, the true parameter values do not lie within the identified interval, different realisations of the same experimental design typically lead to confidence intervals that do contain the true values.”

This statement makes it seem like the authors just do not want to spend time on this example, but frankly this is an important place to make sure everything is working as it should. Moreover it is a nice opportunity to walk the reader through unidentifiability in this problem. Is it true that a larger simulation set will lead to confidence intervals that are not much better? How do we even know how many particles are needed? In all honesty, the authors should consider simulating a -one- layer problem ($L = 70, h = 0.25$ or 0.3) and see how many particles are needed to provide a tight confidence interval. Claims of unidentifiability get stronger when we see what it takes to get positive identifiability.

There also seems to be a missed opportunity to understand the relationship between estimates of the two parameters. When h_1 is high, it stands to reason that h_2 must compensate and be low. But the opposite relation does not seem to hold since the inferred values for h_2 are all low? And why is it that h_1 has such a wide range of viable values, but h_2 does not?

Response: We agree that this problem is of high interest, and we did not mean to skip over this example. We have included new simulations and results in the revised documents to address these questions. Firstly, in the revised Supplementary Material document (Section S1, Figure S1) we repeat the same profiling exercise from Figure 4(a)-(c) except that we show results for 20 identically prepared realizations of the stochastic data. In this suite of simulations we see that the real parameters are often contained within the 95% confidence intervals, and we also see that the profiles are highly variable between different simulation data sets. When we first prepared Figure 4(a)-(c) we made clear note of the fact that the true values were not contained within the identified interval and we wanted to be as clear as possible that this can occur when using a fairly small set of simulations to generate the data. We did not want to hide this fact, and we believe that showing the variability in the revised Supplementary Material document presents a very fair picture of the situation because the profiles can be highly variable when R is relatively small. Showing these highly variable profiles confirms that it is difficult to attribute any meaningful interpretation to the profiles in Figure 4(a)-(c). Indeed, the main take home message here is that when there is insufficient data it is not possible to draw meaningful conclusions, regardless of whether one uses the exact or approximate likelihood function. This is the point that we are making with the results in Figure 4, supported by the additional results in Figure S1.

We would also like to point out that this question prompted us to re-check our results in Figure 4. Since Figure 4 deals with a relatively simple problem with just two parameters, $\theta = (h_1, h_2)$, we evaluated the exact and approximate likelihood functions on a simple two-dimensional of the parameter space and then computed the profile likelihoods by direct column-wise minimisation (i.e. without the MATLAB `fmincon` function). This exercise confirmed that the results in Figure 4 are correct, providing an additional check that the

MATLAB optimization routine converges to the correct result.

Referee 2 asks us a good question about how much data is enough, even for a simple one-layer problem. We address this question in the revised Supplementary Material document (Section 3, Figure S3) where we show that releasing $R = 100$ particles in a simple one-layer problem leads to very accurate estimates of h_1 . This is a useful result because we know that releasing $R = 100$ particles at the end of the second layer in a two-layer problem leads to very wide profiles and poor identifiability. In contrast, when we compare profiles in Figure 4(b)-(c), Figure 4(e)-(f) and Figure 4(h)-(i) we see that the profiles become narrower, and the maximum likelihood estimates of h_1 and h_2 become more accurate as we consider different experimental designs by releasing particles at different locations (as in Figure 4(d)-(f)) and by releasing more particles at multiple locations (as in Figure 4(g)-(i)). Of course, we could continue to release larger numbers of particles at even more locations, but we feel that the results presented in Figure 4 establish the key outcomes. However, we have placed all software on GitHub so that the reader can experiment further with additional experimental designs should they wish.

Referee 2 asks about an interpretation to understand the relationship of the two parameters. We agree that there are some opportunities to explore here, and we felt that this was best explored in the more challenging three-layer problem (Figure 5). Again, we note that when we release $R = 50$ particles at the interface positions in Figure 5 we see that h_1 is well-identified by the data, but that h_2 and h_3 are not. The bivariate profiles in Figure 5(e)-(j) show this since there is a relatively narrow 95% confidence region in the (h_1, h_2) and (h_1, h_3) bivariate profiles indicating that our estimates of h_1 are relatively precise. In contrast, the 95% confidence region in the (h_2, h_3) bivariate profile is very broad, indicating low precision and poor identifiability. These plots show the trade off between our ability to estimate the different parameters with this kind of data.

5. I find the model reduction section at the end somewhat unconvincing. The authors claim that not much is lost when replacing a three layer problem with a two layer problem, but this seems to be because there is no information in either case. The confidence interval for H_2 is [0.224,0.500] (essentially saying it is unidentifiable) as are h_2 and h_3 for the three layer problem. I suspect that if the simulations were greatly increased, at least h_2 would become identifiable and at that point seeing if model reduction is still viable would be meaningful. As of right now, I feel like the statement is “since we don’t have enough data to specify the parameters of a complex model, it is better to have a less complex model that we cannot specify.”

Response: We agree that there is much more that can be done with the model reduction section. Referee 2 is correct to say that the confidence interval for H_2 is wide. We would not say that this is unidentifiable, but rather that it is one-sided unidentifiable. Of course, Referee 2 asks an excellent question about how things change when additional data is used, and new results in the Supplementary Material document (Section S2, Figure S2) show that when we use more particles we obtain increasingly narrow profiles with tighter confidence intervals, and indeed the model reduction is meaningful since H_1 is very close to h_1 whereas the estimate for H_2 interpolates between h_2 and h_3 , as expected. Of course, we could continue to repeat this kind of experiment with increasing numbers of particles, but we feel that releasing our code on GitHub so that the reader can perform these kinds of experiments is a more useful way of addressing this question.

A final point we could like to make is that to further illustrate the value of the model reduction ideas, we have included results for a four-layer problem and a model reduction from four layers to two layers in the revised Supplementary Material document (Supplementary Material, Section S4, Figures S4-S5). These results show that the concept of

model reduction is applicable to more complicated situations with greater numbers of layers. We expect these ideas also hold for problems with greater numbers of layers, and our codes provided on GitHub can be used to explore these ideas further.

6. In spirit I want to like this paper, but the analysis feels incomplete. There seem to be missed opportunities, both in not connecting the model realistically to the data and in not carefully developing the numerical experiments in order to make a clear study of practical and structural identifiability

Response: We thank Referee 2 for this positive closing comment. We agree that all the questions raised by Referee 2 are important and we hope that by adding new data into both the main and online Supplementary Material documents that we have addressed the questions raised by Referee 2. We agree that in the first submission we did not include enough information and this was partly because we were trying to keep within the page limit of *Proceedings A* (20 pages before page charges are levied). However, we hope that by moving some material from the original Supplementary Material document into the main document, and significantly expanding the Supplementary Material document with additional results that Referee 2 feels that the work is now more complete.

Date May 12, 2021
Contact Matthew J. Simpson
Phone +61 4 1369 6607
E-mail matthew.simpson@qut.edu.au
Subject Manuscript Resubmission

Appendix B

Professor Matthew J. Simpson

Professor Matjaz Perč
Editorial Board Member,
Proceedings A

School of Mathematical Sciences
Faculty of Science
Queensland University of Technology
GPO Box 2434, GP Campus
Brisbane, Queensland 4001 Australia

Dear Professor Perč,

Thank you for accepting our manuscript, "Profile likelihood analysis for a stochastic model of diffusion in heterogeneous media", for publication in *Proceedings A*. We are glad to see that Referee 1 had no further suggestions, whereas Referee 2 made two minor comments that we have addressed in the final submission. In addition to addressing these comments we have moved Figure 3 and associated discussion to the revised supplementary material document to ensure that the manuscript is contained within the 20-page limit before page charges are issued.

All changes to the manuscript are highlighted in blue font, and a point-by-point response to each comment is attached. We thank you for your time, and look forward to receiving the proofs.

Yours sincerely,

Matthew J. Simpson

Referee 1

1. The responses from the authors are fine.

Response: We thank Referee 1 for their positive report.

Referee 2

1. The authors have made substantial edits to the paper in response to both the other reviewer and myself. Their response was thorough and, while I still have misgivings about the use of a discretized model in calculating the first-passage time distribution, I do not believe that concern should be a barrier to publication. The new text in the introduction, in particular, does a very nice job of motivating the context and the mode of investigation. Moreover, the new material included in the supplemental information provides further support for their modeling decisions and does inspire confidence that the inference method converges to the correct parameter values (when possible). I would just like to make two notes: neither of which necessitate changes in the text, but I would like to share them with the authors.

Response: We thank Referee 2 for their positive report.

2. First, regarding the use of the discrete model. I certainly understand the convergence of the discrete model to the continuum model in the $\tau \rightarrow 0$ limit. My concern is that the parameter range involves values of h that seem to imply that we are far from the $\tau \rightarrow 0$ limit. Generally, when I think about a random walk model approximating particle based solutions to the heat equation, I am thinking about the random walk being a discretization of a Brownian motion (which can take several steps in a single interval of time). It is only in the $\tau \rightarrow 0$ limit that it becomes unlikely that the Brownian motion would only take one step in the time interval. This is important when using random walks to approximate first-passage time distributions for Brownian motion. By definition, the fastest walkers will be slower than the fastest Brownian motions because they can only take one step at a time

In Figure 2 and 3, the authors provide a plot that intended to show how faithfully simulated fpt properties are captured by the theory. But this is a self-referential consistency between the discrete model and its own estimator. It is still not clear how these compare to the first-passage times of Brownian motions with different diffusivities in different layers. Regardless, the authors do point out that their intent is to say something about whether a FPT method would yield identifiable estimation of parameters; and this investigation can be carried out and be interesting and publishable without fully connecting to the continuum model.

Response: We are glad that Referee 2 is happy with the comparison between simulated first passage time properties and our theory. We fully acknowledge that our discrete mechanism is a commonly-used approach (discrete time, discrete space), but there are other approaches (e.g. continuous time, discrete space or continuous time continuous space). We now fully acknowledge this in the revised manuscript (Page 5, Paragraph 1).

3. Second, the thing that still stands out to me as strange is the way that the method misidentifies the parameters when there are two layers. In the displayed profile likelihoods in Figure 4, the authors share profile likelihoods that exclude the true values from

the presented 95% confidence intervals. The authors call this “unlucky”, at least for the “exact” likelihood profile, which usually includes the true parameter in the 95% range. But looking at Figure S1, the figure 4 plot looks typical for the approximate likelihood profile. Not only are the true values typically excluded from the confidence interval, their values seem to be typically reversed, with h_2 being estimated to have lower values than h_1 . Why would the exact likelihood have modes that are consistent with the real values, but the approximate likelihood have modes that are reversed?

To their credit, the authors do not hide any of this. With the new information in the SI, the reader has enough information to understand what is being done and see when the method is and is not successful, perhaps inspiring other studies.

Response: We are glad Referee 2 appreciates our approach to provide a full, frank and fair assessment of the approximate likelihood. As we point out the approximate likelihood is accurate when we have sufficient data, but the approximation becomes poorer as data becomes scarce. We have mentioned this in several places in the revised manuscript (Page 10, Paragraph 4; Supplementary Material Page 2, Final Paragraph; Supplementary Material Page 3, Paragraph 1; Supplementary Material Page).